# Revisiting Nearest Neighbor for Tabular Data: A Deep Tabular Baseline Two Decades Later

**Han-Jia Ye**[1,2]**, Huai-Hong Yin**[1,2]**, De-Chuan Zhan**[1,2] **& Wei-Lun Chao**[3]
[1]School of Artificial Intelligence, Nanjing University    [2]National Key Laboratory for Novel Software Technology, Nanjing University    [3]The Ohio State University
{yehj, yinhh, zhandc}@lamda.edu.edu.cn, chao.209@osu.edu

## Abstract

The widespread enthusiasm for deep learning has recently expanded into the domain of tabular data. Recognizing that the advancement in deep tabular methods is often inspired by classical methods, *e.g.*, integration of nearest neighbors into neural networks, we investigate whether these classical methods can be revitalized with modern techniques. We revisit a differentiable version of $K$-nearest neighbors (KNN) — Neighbourhood Components Analysis (NCA) — originally designed to learn a linear projection to capture semantic similarities between instances, and seek to gradually add modern deep learning techniques on top. Surprisingly, our implementation of NCA using SGD and without dimensionality reduction already achieves decent performance on tabular data, in contrast to the results of using existing toolboxes like scikit-learn. Further equipping NCA with deep representations and additional training stochasticity significantly enhances its capability, being on par with the leading tree-based method CatBoost and outperforming existing deep tabular models in both classification and regression tasks on 300 datasets. We conclude our paper by analyzing the factors behind these improvements, including loss functions, prediction strategies, and deep architectures. The code is available at https://github.com/LAMDA-Tabular/TALENT.

## 1 Introduction

Tabular data, characterized by its structured format of rows and columns representing individual examples and features, is prevalent in domains like healthcare (Hassan et al., 2020) and e-commerce (Nederstigt et al., 2014). Motivated by the success of deep neural networks in fields like computer vision and natural language processing (Simonyan & Zisserman, 2015; Vaswani et al., 2017; Devlin et al., 2019), numerous deep models have been developed for tabular data to capture complex feature interactions (Cheng et al., 2016; Guo et al., 2017; Popov et al., 2020; Arik & Pfister, 2021; Gorishniy et al., 2021; Katzir et al., 2021; Chang et al., 2022; Chen et al., 2022; Hollmann et al., 2023).

Despite all these attempts, deep tabular models still struggle to match the accuracy of traditional machine learning methods like Gradient Boosting Decision Trees (GBDT) (Prokhorenkova et al., 2018; Chen & Guestrin, 2016) on tabular tasks. Such a fact raises our interest: *to excel in tabular tasks, perhaps deep methods could draw inspiration from traditional methods.* Indeed, several deep tabular methods have demonstrated promising results along this route. Gorishniy et al. (2021); Kadra et al. (2021) consulted classical tabular techniques to design specific MLP architectures and weight regularization strategies, significantly boosting MLPs' accuracy on tabular datasets. Recently, inspired by non-parametric methods (Mohri et al., 2012), TabR (Gorishniy et al., 2024) retrieves neighbors from the entire training set and constructs instance-specific scores with a Transformer-like architecture, leveraging relationships between instances for tabular predictions.

*We follow this route but from a different direction.* Instead of incorporating classic techniques into the already complex deep models, we perform an Occam's-razor-style exploration — starting from the classic method and gradually increasing its complexity by adding modern deep techniques. We hope such an exploration could reveal the key components from both worlds to excel in tabular tasks.

To this end, we build upon TabR (Gorishniy et al., 2024) and choose to start from a classical, differentiable version of $K$-nearest neighbors (KNN) named Neighbourhood Component Analysis

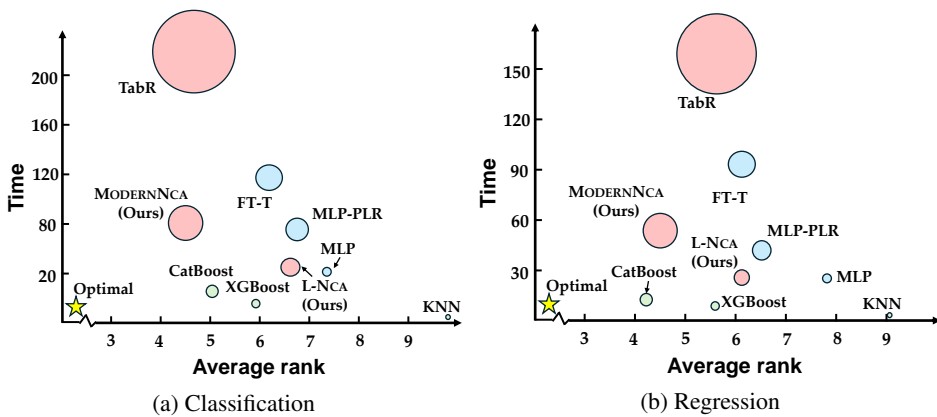

(a) Classification        (b) Regression

Figure 1: Performance-Efficiency-Memory comparison between MODERNNCA and existing methods on classification (a) and regression (b) datasets. Representative tabular prediction methods, including the *classical* methods (in green), the *parametric* deep methods (in blue), and the *non-parametric/neighborhood-based* deep methods (in red), are investigated, based on their records over 300 datasets in Table 1 and Figure 2. The average rank among these eight methods is used as the performance measure. We calculate the average training time (in seconds) and the memory usage of the model (denoted by the radius of the circles, where the larger the circle, the bigger the model). MODERNNCA achieves high training speed compared to other deep tabular models and has a relatively lower memory usage. L-NCA is our *improved linear* version of NCA.

(NCA) (Goldberger et al., 2004). NCA optimizes the KNN prediction accuracy of a target instance by learning a linear projection, ensuring that semantically similar instances are closer than dissimilar ones. Its differentiable nature makes it a suitable backbone for adding deep learning modules.

Our first attempt is to re-implement NCA, using deep learning libraries like PyTorch (Paszke et al., 2019). Interestingly, by replacing the default L-BFGS optimizer (Liu & Nocedal, 1989) in scikit-learn (Pedregosa et al., 2011)[1] with stochastic gradient descent (SGD), we already witnessed a notable accuracy boost on tabular tasks. Further enabling NCA to learn a linear projection into a larger dimensionality (hence not dimensionality reduction) and use a soft nearest neighbor inference rule (Salakhutdinov & Hinton, 2007; Frosst et al., 2019) bring another gain, making NCA on par with deep methods like MLP. (See section 6 for detailed ablation studies and discussions.)

Our second attempt is to replace the linear projection with a neural network for nonlinear embeddings. As NCA's objective function involves the relationship of an instance to all the other training instances, a naive implementation would incur a huge computational burden. We thus employ a stochastic neighborhood sampling (SNS) strategy, randomly selecting a subset of training data as candidate neighbors in each mini-batch. We show that SNS not only improves training efficiency but enhances the model's generalizability, as it introduces additional stochasticity (beyond SGD) in training.

Putting things together, along with the use of a pre-defined feature transform on numerical tabular entries (Gorishniy et al., 2022), our deep NCA implementation, MODERNNCA, achieves remarkably encouraging empirical results. Evaluated on 300 tabular datasets, MODERNNCA is ranked first in classification tasks and just shy of CatBoost (Prokhorenkova et al., 2018) in regression tasks while outperforming other tree-based and deep tabular models. Figure 1 further shows that MODERNNCA well balances training efficiency (with lower training time compared to other deep tabular models), generalizability (with higher average accuracy), and memory efficiency. We also provide a detailed ablation study and discussion on MODERNNCA, comparing different loss functions, training and prediction strategies, and deep architectures, aiming to systematically reveal the impacts of deep learning techniques on NCA, after its release in 2004. In sum, our contributions are two-folded:

- We revisit the classical nearest neighbor approach NCA and systematically explore ways to improve it using modern deep learning techniques.
- Our proposed MODERNNCA achieves outstanding performance in both classification and regression tasks, essentially serving as a strong deep baseline for tabular tasks.

**Remark.** In conducting this study, we become aware of several prior attempts to integrate neural networks into NCA (Salakhutdinov & Hinton, 2007; Min et al., 2010). However, their results and

---

[1] We note that the original NCA paper (Goldberger et al., 2004) did not specify the optimizer.

applicability were downplayed by tree-based methods, and we attribute this to the less powerful deep-learning techniques two decades ago (*e.g.*, restricted Boltzmann machine). In other words, our work can be viewed as a revisit of these attempts from the lens of modern deep-learning techniques.

While our study is largely *empirical*, we believe it offers valuable insights. For years, nearest-neighbor-based methods (though with solid theoretical foundations) have been overlooked in tabular data, primarily due to their low competitiveness with tree-based methods. We hope that our thorough exploration of deep learning techniques for nearest neighbors and the outcome — a strong tabular baseline on par with the leading CatBoost (Prokhorenkova et al., 2018) — would revitalize nearest neighbors and open up new research directions, ideally theoretical foundations behind the improvements.

## 2 RELATED WORK

**Learning with Tabular Data**. Tabular data is a common format across various applications such as click-through rate prediction (Richardson et al., 2007) and time-series forecasting (Ahmed et al., 2010). Tree-based methods like XGBoost (Chen & Guestrin, 2016), LightGBM (Ke et al., 2017), and CatBoost (Prokhorenkova et al., 2018) have proven effective at capturing feature interactions and are widely used in real-world applications. Recognizing the ability of deep neural networks to learn feature representations from raw data and make nonlinear predictions, recent methods have applied deep learning techniques to tabular models (Cheng et al., 2016; Guo et al., 2017; Popov et al., 2020; Borisov et al., 2022; Arik & Pfister, 2021; Kadra et al., 2021; Katzir et al., 2021; Chen et al., 2022; Zhou et al., 2023). For instance, deep architectures such as residual networks and transformers have been adapted for tabular prediction (Gorishniy et al., 2021; Hollmann et al., 2023). Moreover, data augmentation strategies have been introduced to mitigate overfitting in deep models (Ucar et al., 2021; Bahri et al., 2022; Rubachev et al., 2022). Deep tabular models have demonstrated competitive performance across a wide range of applications. However, researchers have observed that deep models still face challenges in capturing high-order feature interactions as effectively as tree-based models (Grinsztajn et al., 2022; McElfresh et al., 2023; Ye et al., 2024a).

**NCA Variants**. Nearest Neighbor approaches make predictions based on the relationships between an instance and its neighbors in the training set. Instead of identifying neighbors using raw features, NCA employs a differentiable Nearest Neighbor loss function (also known as soft-NN loss) to learn a linear projection for better distance measurement (Goldberger et al., 2004). Several works have extended this idea with alternative loss functions (Globerson & Roweis, 2005; Tarlow et al., 2013), while others explore NCA variants for data visualization (Venna et al., 2010). A few nonlinear extensions of NCA, developed over a decade ago, demonstrated a bit improved performance on image classification tasks using architecture like restricted Boltzmann machines (Salakhutdinov & Hinton, 2007; Min et al., 2010). For visual tasks, the entanglement effects of soft-NN loss on deep learned representations have been analyzed (Frosst et al., 2019), and variants of this loss have been applied to few-shot learning scenarios (Vinyals et al., 2016; Laenen & Bertinetto, 2021). The effectiveness of NCA variants in fields like image recognition suggests untapped potential (Wu et al., 2018), motivating our revisit of this method with modern deep learning techniques for tabular data.

**Metric Learning**. NCA is a form of metric learning (Xing et al., 2002), where a projection is learned to pull similar instances closer together and push dissimilar ones farther apart, leading to improved classification and regression performance with KNN (Davis et al., 2007; Weinberger & Saul, 2009; Kulis, 2013; Bellet et al., 2015; Ye et al., 2020). Initially applied to tabular data, metric learning has evolved into a valuable tool, particularly when integrated with deep learning techniques, across domains like image recognition (Schroff et al., 2015; Sohn, 2016; Song et al., 2016; Khosla et al., 2020), person re-identification (Yi et al., 2014; Yang et al., 2018), and recommendation systems (Hsieh et al., 2017; Wei et al., 2023). Recently, LocalPFN (Thomas et al., 2024) incorporates KNN with TabPFN. TabR (Gorishniy et al., 2024) introduced a feed-forward network with a custom attention-like mechanism to retrieve neighbors for each instance, enhancing tabular prediction tasks. Despite its promising results, the high computational cost of neighborhood selection and the complexity of its architecture limit the practicality of TabR. In contrast, our paper revisits NCA and proposes a simpler deep tabular baseline that maintains efficient training speeds without sacrificing performance.

## 3 PRELIMINARY

In this section, we first introduce the task learning with tabular data. We then provide a brief overview of NCA (Goldberger et al., 2004) and TabR (Gorishniy et al., 2024).

### 3.1 LEARNING WITH TABULAR DATA

A labeled tabular dataset is formatted as $N$ examples (rows in the table) and $d$ features/attributes (columns in the table). An instance $\boldsymbol{x}_i$ is depicted by its $d$ feature values. There are two kinds of features: the numerical (continuous) ones and categorical (discrete) ones. Given $x_{i,j}$ as the $j$-th feature of instance $\boldsymbol{x}_i$, we use $x_{i,j}^{\text{num}} \in \mathbb{R}$ and $\boldsymbol{x}_{i,j}^{\text{cat}}$ to denote numerical (*e.g.*, the height of a person) and categorical (*e.g.*, the gender of a person) feature values of an instance, respectively. The categorical features are usually transformed in a one-hot manner, *i.e.*, $\boldsymbol{x}_{i,j}^{\text{cat}} \in \{0,1\}^{K_j}$, where the index of value 1 indicates the category among the $K_j$ options. We assume the instance $\boldsymbol{x}_i \in \mathbb{R}^d$ w.l.o.g. and will explore other encoding strategies later. Each instance is associated with a label $y_i$, where $y_i \in [C] = \{1, \ldots, C\}$ in a multi-class classification task and $y_i \in \mathbb{R}$ in a regression task.

Given a tabular dataset $\mathcal{D} = \{(\boldsymbol{x}_i, y_i)\}_{i=1}^N$, we aim to learn a model $f$ on $\mathcal{D}$ that maps $\boldsymbol{x}_i$ to its label $y_i$. We measure the quality of $f$ by the joint likelihood over $\mathcal{D}$, *i.e.*, $\max_f \prod_{(\boldsymbol{x}_i, y_i) \in \mathcal{D}} \Pr(y_i \mid f(\boldsymbol{x}_i))$. The objective could be reformulated in the form of negative log-likelihood of the true labels,

$$\min_f \sum_{(\boldsymbol{x}_i, y_i) \in \mathcal{D}} -\log \Pr(y_i \mid f(\boldsymbol{x}_i)) = \sum_{(\boldsymbol{x}_i, y_i) \in \mathcal{D}} \ell(y_i, \hat{y}_i = f(\boldsymbol{x}_i)) , \quad (1)$$

or equivalently, the discrepancy between the predicted label $\hat{y}_i$ and the true label $y_i$ measured by the loss $\ell(\cdot, \cdot)$, *e.g.*, cross-entropy. We expect the learned model $f$ is able to extend its ability to unseen instances sampled from the same distribution as $\mathcal{D}$. $f$ could be implemented with classical methods such as SVM and tree-based approaches or MLPs.

### 3.2 NEAREST NEIGHBOR FOR TABULAR DATA

**KNN** is one of the most representative non-parametric tabular models for classification and regression — making predictions based on the labels of the nearest neighbors (Bishop, 2006; Mohri et al., 2012). In other words, the prediction $f(\boldsymbol{x}_i; \mathcal{D})$ of the model $f$ conditions on the whole training set. Given an instance $\boldsymbol{x}_i$, KNN calculates the distance between $\boldsymbol{x}_i$ and other instances in $\mathcal{D}$. Assume the $K$ nearest neighbors are $\mathcal{N}(\boldsymbol{x}_i; \mathcal{D}) = \{(\boldsymbol{x}_1, y_1), \ldots, (\boldsymbol{x}_K, y_K)\}$, then, the label $y_i$ of $\boldsymbol{x}_i$ is predicted based on those labels in the neighbor set $\mathcal{N}(\boldsymbol{x}_i; \mathcal{D})$. For classification task $\hat{y}_i$ is the majority voting of labels in $\mathcal{N}(\boldsymbol{x}_i; \mathcal{D})$ while is the average of those labels in regression tasks.

The distance $\text{dist}(\boldsymbol{x}_i, \boldsymbol{x}_j)$ in KNN determines the set of nearest neighbors $\mathcal{N}(\boldsymbol{x}_i; \mathcal{D})$, which is one of its key factors. The Euclidean distance between a pair $(\boldsymbol{x}_i, \boldsymbol{x}_j)$ is $\text{dist}(\boldsymbol{x}_i, \boldsymbol{x}_j) = \sqrt{(\boldsymbol{x}_i - \boldsymbol{x}_j)^\top (\boldsymbol{x}_i - \boldsymbol{x}_j)}$. A distance metric that reveals the characteristics of the dataset will improve KNN and lead to more accurate predictions (Xing et al., 2002; Davis et al., 2007; Weinberger & Saul, 2009; Bellet et al., 2015).

**Neighbourhood Component Analysis (NCA).** NCA focuses on the classification task (Goldberger et al., 2004). According to the 1NN rule, NCA defines the probability that $\boldsymbol{x}_j$ locates in the neighborhood of $\boldsymbol{x}_i$ by

$$\Pr(\boldsymbol{x}_j \in \mathcal{N}(\boldsymbol{x}_i; \mathcal{D}) \mid \boldsymbol{x}_i, \mathcal{D}, \boldsymbol{L}) = \frac{\exp\left(-\text{dist}^2(\boldsymbol{L}^\top \boldsymbol{x}_i, \boldsymbol{L}^\top \boldsymbol{x}_j)\right)}{\sum_{(\boldsymbol{x}_l, y_l) \in \mathcal{D}, \boldsymbol{x}_l \neq \boldsymbol{x}_i} \exp\left(-\text{dist}^2(\boldsymbol{L}^\top \boldsymbol{x}_i, \boldsymbol{L}^\top \boldsymbol{x}_l)\right)} . \quad (2)$$

Then, the posterior probability that an instance $\boldsymbol{x}_i$ is classified as the class $y_i$ is:

$$\Pr(\hat{y}_i = y_i \mid \boldsymbol{x}_i, \mathcal{D}, \boldsymbol{L}) = \sum_{(\boldsymbol{x}_j, y_j) \in \mathcal{D} \wedge y_j = y_i} \Pr(\boldsymbol{x}_j \in \mathcal{N}(\boldsymbol{x}_i; \mathcal{D}) \mid \boldsymbol{x}_i, \mathcal{D}, \boldsymbol{L}) . \quad (3)$$

$\boldsymbol{L} \in \mathbb{R}^{d \times d'}$ is a linear projection usually with $d' \leq d$, which reduces the dimension of the raw input. Therefore, the posterior that an instance $\boldsymbol{x}_i$ belongs to the class $y_i$ depends on its similarity (measured by the negative squared Euclidean distance in the space projected by $\boldsymbol{L}$) between its neighbors from

class $y_i$ in $\mathcal{D}$. Equation 3 approximates the expected leave-one-out error for $\boldsymbol{x}_i$, and the original NCA maximizes the **sum of** $\Pr(\hat{y}_i = y_i \mid \boldsymbol{x}_i, \mathcal{D}, \boldsymbol{L})$ over all instances in $\mathcal{D}$. Instead of considering all instances in the neighborhood equally, this objective mimics a soft version of KNN, where all instances in the training set are weighted (nearer neighbors have more weight) for the nearest neighbor decision. In the test stage, KNN is applied to classify an unseen instance in the space projected by $\boldsymbol{L}$.

**TabR** is a deep tabular method that retrieves the neighbors of an instance $\boldsymbol{x}_i$ using deep neural networks. Specifically, TabR identifies the $K$ nearest neighbors in the embedding space and defines the contribution of each neighbor $(\boldsymbol{x}_j, y_j)$ to $\boldsymbol{x}_i$ as follows: $s(\boldsymbol{x}_i, \boldsymbol{x}_j, y_j) = \boldsymbol{W}\boldsymbol{y}_j + \mathrm{T}(\boldsymbol{L}^\top E(\boldsymbol{x}_j) - \boldsymbol{L}^\top E(\boldsymbol{x}_i))$. Here, $\mathrm{T}$ is a transformation composed of a linear layer without bias, dropout, ReLU activation, and another linear layer. $E$ represents the encoder module for TabR, while $\boldsymbol{W}$ is a linear projection and $\boldsymbol{y}_j$ is the encoded label vector of $y_j$. The instance-specific scores are then aggregated as: $R(\boldsymbol{x}_j, y_j, \boldsymbol{x}_i) = \sum_{(\boldsymbol{x}_j, y_j) \in \mathcal{D}} \alpha_j \cdot s(\boldsymbol{x}_i, \boldsymbol{x}_j, y_j)$, where the weight $\alpha_j$ is defined as $\alpha_j \propto \{-\operatorname{dist}(\boldsymbol{L}^\top E(\boldsymbol{x}_j), \ \boldsymbol{L}^\top E(\boldsymbol{x}_i))\}$ and normalized using a softmax function. Finally, $R(\boldsymbol{x}_j, y_j, \boldsymbol{x}_i)$ is added to $E(\boldsymbol{x}_i)$, and the result is processed by a prediction module to obtain $\hat{y}_i$. For further details, including instance-level layer normalization, numerical attribute encoding, and the selection strategy for $K$ nearest neighbors in the summation, please refer to Gorishniy et al. (2024).

## 4    MODERNNCA

Given the promising results of TabR on tabular data, we take the original NCA as our starting point and gradually enhance its complexity by incorporating modern deep learning techniques. This Occam's-razor-style exploration may allow us to identify the key components that lead to strong performance in tabular tasks, drawing insights from both classical and deep tabular models. In the following, we introduce our proposed MODERNNCA (abbreviated as M-NCA) through two key attempts to improve upon the original NCA.

### 4.1    THE FIRST ATTEMPT

We generalize the projection in Equation 2 by introducing a transformation $\phi$, which maps $\boldsymbol{x}_i$ into a space with dimensionality $d'$. To remain consistent with the original NCA, we initially define $\phi$ as a linear layer, *i.e.*, $\phi(\boldsymbol{x}_i) = \operatorname{Linear}(\boldsymbol{x}_i)$, consisting of a linear projection and a bias term.

**Learning Objective**. Assume the label $y_j$ is continuous in regression tasks and in one-hot form for classification tasks. We modify Equation 3 as follows:

$$\hat{y}_i = \sum_{(\boldsymbol{x}_j, y_j) \in \mathcal{D}} \frac{\exp\left(-\operatorname{dist}^2(\phi(\boldsymbol{x}_i), \ \phi(\boldsymbol{x}_j))\right)}{\sum_{(\boldsymbol{x}_l, y_l) \in \mathcal{D}, \boldsymbol{x}_l \neq \boldsymbol{x}_i} \exp\left(-\operatorname{dist}^2(\phi(\boldsymbol{x}_i), \ \phi(\boldsymbol{x}_l))\right)} y_j \ . \tag{4}$$

This formulation ensures that similar instances (based on their distance in the embedding space mapped by $\phi$) yield closer predictions. For classification, Equation 4 generalizes Equation 3, predicting the label of a target instance by computing a weighted average of its neighbors across the $C$ classes. Here, $\hat{y}_i \in \mathbb{R}^C$ is a probability vector representing $\{\Pr(\hat{y}_i = c \mid \boldsymbol{x}_i, \mathcal{D}, \phi)\}_{c \in [C]}$. In regression tasks, the prediction is the weighted sum of scalar labels from the neighborhood.

By combining Equation 3 with Equation 1, we define $\ell$ in Equation 1 as negative log-likelihood for classification and mean square error for regression. This classification loss is also known as the soft Nearest Neighbor (soft-NN) loss (Frosst et al., 2019; Khosla et al., 2020) for visual tasks. Different from Goldberger et al. (2004); Salakhutdinov & Hinton (2007) that used **sum of probability** as in the original NCA's loss, we find **sum of log probability** provides better performance on tabular data.

**Prediction Strategy**. For a test instance, the original NCA projects all instances using the learned $\phi$ and applies KNN to classify the test instance based on its neighbors from the entire training set $\mathcal{D}$. Instead of employing the traditional "hard" KNN approach, we adopt the soft-NN rule (Equation 4) to estimate the label posterior, applicable to both classification and regression. Specifically, in the classification case, Equation 4 produces a $C$-dimensional vector, with the index of the maximum value indicating the predicted class. For regression, $\hat{y}_i$ directly corresponds to the predicted value.

Furthermore, we do not limit the mapping to dimensionality reduction. The linear projection $\phi$ can transform $\boldsymbol{x}_i$ into a higher-dimensional space if necessary. We also replace the L-BFGS optimizer (used in scikit-learn) with stochastic gradient descent (SGD) for better scalability and performance.

These modifications result in a notable accuracy boost for NCA on tabular tasks, making it competitive with deep models like MLP. We refer to this improved version of (linear) NCA as L-NCA.

## 4.2 THE SECOND ATTEMPT

We further enhance L-NCA by incorporating modern deep learning techniques, leading to our strong deep tabular baseline, MODERNNCA (M-NCA).

**Architectures.** To introduce nonlinearity into the model, we first enhance the transformation $\phi$ in subsection 4.1 with multiple nonlinear layers appended. Specifically, we define a one-layer nonlinear mapping as a sequence of operators following Gorishniy et al. (2021), consisting of one-dimensional batch normalization (Ioffe & Szegedy, 2015), a linear layer, ReLU activation, dropout (Srivastava et al., 2014), and another linear layer. In other words, the input $\boldsymbol{x}_i$ will be transformed by

$$g(\boldsymbol{x}_i) = \text{Linear}\left(\text{Dropout}\left(\left(\text{ReLU}\left(\text{Linear}\left(\text{BatchNorm}\left(\boldsymbol{x}_i\right)\right)\right)\right)\right)\right) . \tag{5}$$

One or more layers of such a block $g$ can be appended on top of the original linear layer in subsection 4.1 to implement the final nonlinear mapping $\phi$, which further incorporates an additional batch normalization at the end to calibrate the output embedding. Empirical results show that batch normalization outperforms other normalization strategies, such as layer normalization (Ba et al., 2016), in learning a robust latent embedding space.

For categorical input features, we use one-hot encoding, and for numerical features, we leverage PLR (lite) encoding, following TabR (Gorishniy et al., 2024). PLR encoding combines periodic embeddings, a linear layer, and ReLU to project instances into a high-dimensional space, thereby increasing the model's capacity with additional nonlinearity (Gorishniy et al., 2022). PLR (lite) restricts the linear layer to be shared across all features, balancing complexity and efficiency.

**Stochastic Neighborhood Sampling**. SGD is commonly applied to optimize deep neural networks — a mini-batch of instances is sampled, and the average instance-wise loss in the mini-batch is calculated for back-propagation. However, the instance-wise loss based on the predicted label in Equation 4 involves pairwise distances between an instance in the mini-batch and the entire training set $\mathcal{D}$, imposing a significant computational burden.

To accelerate the training speed of MODERNNCA, we propose a Stochastic Neighborhood Sampling (SNS) strategy. In SNS, a subset $\hat{\mathcal{D}}$ of the training set $\mathcal{D}$ is randomly sampled for each mini-batch, and only distances between instances in the mini-batch and this subset are calculated. In other words, $\hat{\mathcal{D}}$ replaces $\mathcal{D}$ in Equation 4, and only the labels in $\hat{\mathcal{D}}$ are used to predict the label of a given instance during training. During inference, however, the model resumes the searches for neighbors using the entire training set $\mathcal{D}$. Unlike deep metric learning methods that only consider pairs of instances within a sampled mini-batch (Schroff et al., 2015; Song et al., 2016; Sohn, 2016), *i.e.*, $\hat{\mathcal{D}}$ is the mini-batch, our SNS approach retains both efficiency and diversity in the selection of neighbor candidates.

We empirically observed that SNS not only increases the training efficiency of MODERNNCA, since fewer examples are utilized for back-propagation, but also improves the generalization ability of the learned mapping $\phi$. We attribute the improvement to the fact that $\phi$ is learned on more difficult, stochastic prediction tasks. The resulting $\phi$ thus becomes more robust to the potentially noisy and unstable neighborhoods in the test scenario. The influence of sampling ratio and other sampling strategies are investigated in detail in the experiments.

**Distance Function.** Empirically, we find that using the Euclidean distance instead of its squared form in Equation 4 leads to further performance improvements. Therefore, we adopt Euclidean distance as the default. Comparisons of various distance functions are provided in the appendix.

## 5 EXPERIMENTS

### 5.1 SETUPS

We evaluate MODERNNCA on 300 datasets from a recently released large-scale tabular benchmark (Ye et al., 2024a), comprising 120 binary classification datasets, 80 multi-class classification datasets, and 100 regression datasets sourced from UCI, OpenML (Vanschoren et al., 2014), Kaggle,

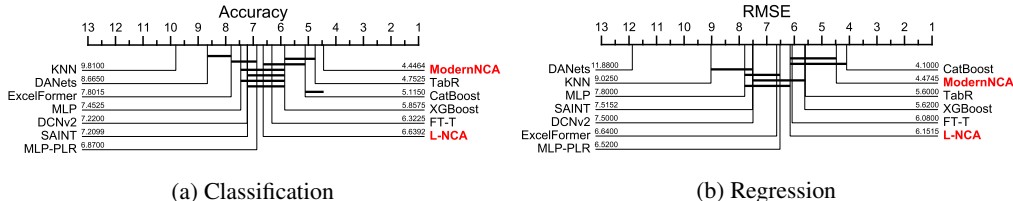

(a) Classification              (b) Regression

Figure 2: The critical difference diagrams based on the Wilcoxon-Holm test with a significance level of 0.05 to detect pairwise significance for both classification tasks (evaluated using accuracy) and regression tasks (evaluated using RMSE).

Table 1: The Win/Tie/Lose ratio between MODERNNCA and 20 comparison methods across the 300 datasets, covering both classification (based on accuracy) and regression tasks (based on RMSE). This ratio is determined using a significant $t$-test at a 95% confidence interval.

| Method | Win | Tie | Lose | Method | Win | Tie | Lose |
|---|---|---|---|---|---|---|---|
| SVM | 0.78 | 0.13 | 0.10 | KNN | 0.79 | 0.07 | 0.14 |
| SwitchTab (Wu et al., 2024) | 0.88 | 0.09 | 0.03 | DANets (Chen et al., 2022) | 0.74 | 0.18 | 0.08 |
| NODE (Popov et al., 2020) | 0.70 | 0.15 | 0.15 | Tangos (Jeffares et al., 2023) | 0.66 | 0.20 | 0.14 |
| TabCaps (Chen et al., 2023) | 0.64 | 0.23 | 0.13 | PTaRL (Ye et al., 2024b) | 0.62 | 0.22 | 0.16 |
| DCNv2 (Wang et al., 2021) | 0.62 | 0.20 | 0.18 | MLP (Gorishniy et al., 2021) | 0.61 | 0.23 | 0.15 |
| ResNet (Gorishniy et al., 2021) | 0.59 | 0.30 | 0.11 | MLP-PLR (Gorishniy et al., 2022) | 0.57 | 0.27 | 0.16 |
| RandomForest | 0.57 | 0.18 | 0.26 | ExcelFormer (Chen et al., 2024) | 0.56 | 0.28 | 0.16 |
| SAINT (Somepalli et al., 2022) | 0.55 | 0.28 | 0.18 | FT-T (Gorishniy et al., 2021) | 0.50 | 0.28 | 0.23 |
| XGBoost (Chen & Guestrin, 2016) | 0.49 | 0.19 | 0.32 | LightGBM Ke et al. (2017) | 0.45 | 0.23 | 0.32 |
| TabR (Gorishniy et al., 2024) | 0.41 | 0.36 | 0.23 | CatBoost (Prokhorenkova et al., 2018) | 0.38 | 0.27 | 0.35 |

and other repositories. The dataset collection in Ye et al. (2024a) was carefully curated, considering factors such as data diversity, representativeness, and quality mentioned in Kohli et al. (2024); Tschalzev et al. (2024).

**Evaluation.** We follow the evaluation protocol from Gorishniy et al. (2021; 2024). Each dataset is randomly split into training, validation, and test sets in proportions of 64%/16%/20%, respectively. For each dataset, we train each model using 15 different random seeds and calculate the average performance on the test set. For classification tasks, we consider accuracy (higher is better), and for regression tasks, we use Root Mean Square Error (RMSE) (lower is better). To summarize overall model performance, we report the average performance rank across all methods and datasets (lower ranks are better), following Delgado et al. (2014); McElfresh et al. (2023). Additionally, we conduct statistical $t$-tests to determine whether the differences between MODERNNCA and other methods are statistically significant.

**Comparison Methods.** We compare MODERNNCA with 20 approaches among three different categories, including classical parametric methods, parametric deep models, and neighborhood-based methods. For brevity, only 8 of them are shown in Figure 1.

**Implementation Details.** We pre-process all datasets following Gorishniy et al. (2021). For all deep methods, we set the batch size as 1024. The hyper-parameters of all methods are searched based on the training and validation set via Optuna (Akiba et al., 2019) following Gorishniy et al. (2021; 2024) over 100 trials. We set the ranges of the hyper-parameters for the compared methods following Gorishniy et al. (2021; 2024) and their official codes. The best-performed hyper-parameters are fixed during the final 15 seeds. Since the sampling rate of SNS effectively enhances the performance and reduces the training overhead, we treat it as a hyper-parameter and search within the range of [0.05, 0.6]. For additional implementation details, please refer to Liu et al. (2024).

## 5.2 MAIN RESULTS

The comparison results between MODERNNCA, L-NCA, and six representative methods are presented in Figure 1. All methods are evaluated across three aspects: performance (average performance rank), average training time, and average memory usage across all datasets. While some models,

such as TabR, exhibit strong performance, they require significantly longer training times. In contrast, MODERNNCA strikes an excellent balance across various evaluation criteria.

We also applied the Wilcoxon-Holm test (Demsar, 2006) to assess pairwise significance among all methods for both classification and regression tasks. The results are shown in Figure 2. For classification tasks (shown in the left part of Figure 2), MODERNNCA consistently outperforms tree-based methods like XGBoost in most cases, demonstrating that its deep neural network architecture is more effective at capturing nonlinear relationships. Furthermore, compared to deep tabular models such as FT-T and MLP-PLR, MODERNNCA maintains its superiority. When combined with the results in Figure 1, these observations validate the effectiveness of MODERNNCA. It achieves performance on par with the leading tree-based method, CatBoost, while outperforming existing deep tabular models in both classification and regression tasks across 300 datasets.

Additionally, we calculated the Win/Tie/Lose ratio between MODERNNCA and other comparison methods across the 300 datasets. If two methods show no significant difference (based on a $t$-test at a 95% confidence interval), they are considered tied. Otherwise, one method is declared the winner based on the comparison of their average performance. Given the no free lunch theorem, it is challenging for any single method to statistically outperform others across all cases. Nevertheless, MODERNNCA demonstrates superior performance in most cases. For instance, MODERNNCA outperforms TabR on 123 datasets, ties on 108 datasets, and does so with a simpler architecture and shorter training time. Compared to CatBoost, MODERNNCA wins on 114 datasets and ties on 81 datasets. These results indicate that MODERNNCA serves as an effective and competitive deep learning baseline for tabular data.

## 6 ANALYSES AND ABLATION STUDIES OF MODERNNCA

In this section, we analyze the sources of improvement in MODERNNCA. All experiments are conducted on a tiny tabular benchmark comprising 45 datasets, as introduced in (Ye et al., 2024a). The benchmark consists of 27 classification datasets and 18 regression datasets. The average rank of various tabular methods on this benchmark closely aligns with the results observed on the larger set of 300 datasets, as detailed in (Ye et al., 2024a).

### 6.1 IMPROVEMENTS FROM NCA TO L-NCA

We begin with the original NCA (Goldberger et al., 2004), using the scikit-learn implementation (Pedregosa et al., 2011). We progressively replace key components in NCA and assess the resulting performance improvements. Since the original NCA only targets classification tasks, this subsection focuses on the 27 classification datasets in the tiny benchmark. To ensure a fair comparison, we re-implement the original NCA using the deep learning framework PyTorch (Paszke et al., 2019), denoting this baseline version as "NCAv0".

**Does Projection to a Higher Dimension Help?** In the scikit-learn implementation, NCA is constrained to perform dimensionality reduction, *i.e.*, $d' \leq d$ for the projection $\boldsymbol{L}$. We remove this constraint, allowing NCA to project into higher dimensions, and refer to this version as "NCAv1". Although higher dimensions by linear projections do not inherently enhance the representation ability of the squared Euclidean distance, the improvements in average performance rank from NCAv0 to NCAv1 (shown in Table 2) indicate that projecting to a higher dimension facilitates the optimization of this non-convex problem and improves generalization.

**Does Stochastic Gradient Descent Help?** Stochastic gradient descent (SGD) is a widely used optimizer in deep learning. To explore whether SGD can improve NCA's performance, we replace the default L-BFGS optimizer used in scikit-learn with SGD (without momentum) and denote this variant as "NCAv2". The performance improvements from NCAv1 to NCAv2 in Table 2 indicate that SGD makes NCA more effective in tabular data tasks.

**The Influence of the Loss Function.** The original NCA maximizes the expected leave-one-out accuracy as shown in Equation 3. In contrast, we minimize the negative log version of this objective as described in Equation 1. Although the log version for classification tasks was mentioned in Goldberger et al. (2004); Salakhutdinov & Hinton (2007), the original NCA preferred the leave-one-out formulation for better performance. We denote the variant with the modified loss function as

Table 2: Comparison of the average rank of (the linear) NCA variants and (the nonlinear) MLP across 27 classification datasets in the tiny-benchmark. The check marks indicate the differences in components among the variants. The average rank represents the overall performance of a method across all datasets, with lower ranks indicating better performance. The final variant, NCAv4, corresponds to the L-NCA version discussed in our paper.

| | High dimension | SGD optimizer | Log loss | Soft-NN prediction | Average rank |
|---|---|---|---|---|---|
| NCAv0 | | | | | 4.400 |
| NCAv1 | ✓ | | | | 3.708 |
| NCAv2 | ✓ | ✓ | | | 3.296 |
| NCAv3 | ✓ | ✓ | ✓ | | 3.192 |
| NCAv4 | ✓ | ✓ | ✓ | ✓ | 2.962 |
| MLP | ✓ | ✓ | ✓ | | 3.000 |

Table 3: Comparison among various configurations of the deep architectures used to implement $\phi$, where MLP is the default choice in MODERNNCA. We show the change in average performance rank (lower is better) across the four configurations on the 45 datasets in the tiny benchmark.

| | MLP | Linear | w/ LayerNorm | ResNet |
|---|---|---|---|---|
| Classification | 2.333 | 2.778 | 2.537 | 2.352 |
| Regression | 2.333 | 2.433 | 2.528 | 2.806 |

"NCAv3". As shown in Table 2 (NCAv2 vs. NCAv3), we find that using the log version slightly improves performance, especially when combined with deep architectures used in MODERNNCA. Further comparisons with alternative objectives are provided in the appendix.

**The Influence of the Prediction Strategy.** During testing, rather than applying a "hard" KNN with the learned embeddings as in standard metric learning, we adopt a soft nearest neighbor (soft-NN) inference rule, consistent with the training phase. This variant, using soft-NN for prediction, is referred to as "NCAv4", which is equivalent to the "L-NCA" version defined in subsection 4.1. Based on the change of average performance rank in Table 2, this modified prediction strategy further enhances NCA's classification performance, bringing linear NCA surpassing deep models like MLP.

## 6.2 IMPROVEMENTS FROM L-NCA TO M-NCA

In this subsection, we investigate the influence of architectures and encoding strategies to systematically reveal the impacts of more deep learning techniques on NCA.

**Linear vs. Deep Architectures**. We first investigate the architecture design for $\phi$ in MODERNNCA, where one or more layers of blocks $g(\cdot)$ are added on top of a linear projection. We consider three configurations. First, we set $\phi$ as a linear projection, where the dimensionality of the projected space is a hyper-parameter.[2] Then we replace batch normalization with layer normalization in the block. Finally, we add a residual link from the block's input to its output. Based on classification and regression performance across 45 datasets, we present the average performance rank of the four variants in Table 3. To avoid limiting model capacity, hyper-parameters such as the number of layers are determined based on the validation set. Further comparisons of fixed architecture configurations are listed in the appendix.

We first compare NCA with MLP vs. with the linear counterpart in Table 3. In classification tasks, MLP achieves a lower rank, highlighting the importance of incorporating nonlinearity into the model. However, in regression tasks, the linear version performs well, with MLP showing only small improvements. Although the linear projection is part of MLP's search space, the linear version benefits from a smaller hyper-parameter space, potentially resulting in better generalization.

As described in subsection 4.2, MLP uses batch normalization instead of layer normalization. Empirically, batch normalization performs better on average in both classification and regression tasks as shown in Table 3. Additionally, we compare the MLP implementation with and without residual connections. While performing similarly in classification, MLP shows superiority, especially in regression. Therefore, we adopt the MLP implementation in Table 3 for MODERNNCA.

---

[2]This "linear" version also includes the SNS sampling strategy and the nonlinear PLR encoding.

Table 4: Comparison among MODERNNCA, MLP (Gorishniy et al., 2021), and TabR (Gorishniy et al., 2024) with or without PLR encoding for numerical features. We show the change in average performance rank across the four configurations on the 45 datasets in the tiny-benchmark.

| | w/o PLR | | | w/ PLR | | |
|---|---|---|---|---|---|---|
| | MLP | TabR | MODERNNCA | MLP | TabR | MODERNNCA |
| Classification | 4.556 | 3.148 | 3.037 | 4.480 | 3.037 | 2.630 |
| Regression | 4.444 | 3.167 | 3.389 | 3.333 | 3.444 | 3.222 |

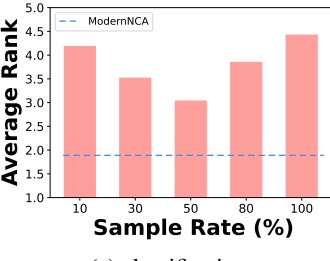
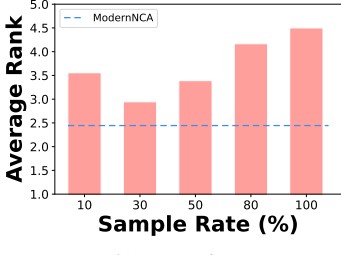

(a) classification    (b) regression

Figure 3: The change of average performance rank with different sampling rates among {10%, 30%, 50%, 80%, 100%} in SNS strategy. The dotted line denotes the rank of MODERNNCA.

**Influence of the PLR Encoding**. PLR encoding transforms numerical features into high-dimensional vectors, enhancing both model capacity and nonlinearity. To assess the impact of PLR encoding, we compare MODERNNCA with MLP and TabR, both with and without PLR encoding. Following a similar setup as in Table 3, we present the change in average performance rank across six methods in both classification and regression tasks in Table 4.

Without PLR encoding, TabR outperforms MLP, and MODERNNCA shows stronger performance in classification while performing slightly worse in regression (although still better than MLP). PLR encoding improves all methods, as evidenced by the decrease in average performance rank. In the right section of Table 4, we observe that MODERNNCA performs best in both classification and regression tasks, effectively leveraging PLR encoding better than TabR. This may be because the nonlinearity introduced by PLR compensates for the relative simplicity of MODERNNCA. The results also validate that the strength of MODERNNCA comes from a combination of its objective, architecture, and training strategy, rather than relying solely on the PLR encoding strategy.

**The Influence of Sampling Ratios**. Due to the huge computational cost of calculating distances in the learned embedding space, MODERNNCA employs a Stochastic Neighborhood Sampling (SNS) strategy, where only a subset of the training data is randomly sampled for each mini-batch.Therefore, the training time and memory cost is significantly reduced. We experiment with varying the proportion of sampled training data while keeping other hyper-parameters constant, then evaluate the corresponding test performance. As shown in Figure 3, sampling 30%-50% of the training set yields better results for MODERNNCA than using the full set. SNS not only improves training efficiency but also enhances the model's generalization ability. The plots also indicate that, with a tuned sampling ratio, MODERNNCA achieves a superior performance rank (dotted lines in the figure).

# 7    CONCLUSION

Leveraging neighborhood relationships for predictions is a classical approach in machine learning. In this paper, we revisit and enhance one of the most representative neighborhood-based methods, NCA, by incorporating modern deep learning techniques. The improved MODERNNCA establishes itself as a strong baseline for deep tabular prediction tasks, offering competitive performance while reducing the training time required to access the entire dataset. Extensive results demonstrate that MODERNNCA frequently outperforms both tree-based and deep tabular models in classification and regression tasks. Our detailed analyses shed light on the key factors driving these improvements, including the enhancements introduced to the original NCA.

ACKNOWLEDGMENT

This research is partially supported by NSFC (62376118), Collaborative Innovation Center of Novel Software Technology and Industrialization, Key Program of Jiangsu Science Foundation (BK20243012). We thank Si-Yang Liu and Hao-Run Cai for helpful discussions.

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

The Appendix consists of two sections:

- Appendix A: Datasets and implementation details.
- Appendix B: Additional experimental results.

## APPENDIX A    DATASETS AND IMPLEMENTATION DETAILS

In this section, we outline the preprocessing steps applied to the datasets before training, as well as descriptions of the datasets used.

### A.1    DATA PRE-PROCESSING

We follow the data preprocessing pipeline from Gorishniy et al. (2021) for all methods. For numerical features, we apply standardization by subtracting the mean and scaling the values. For categorical features, we use one-hot encoding to convert them for model input.

### A.2    DATASET INFORMATION

We use the recent large-scale tabular benchmark from Ye et al. (2024a), which includes 300 datasets covering various domains such as healthcare, biology, finance, education, and physics. The dataset sizes range from 1,000 to 1 million instances. More detailed information on the datasets can be found in Ye et al. (2024a).

For each dataset, we randomly sample 20% of the instances to form the test set. The remaining 80% is split further, with 20% of which held out as a validation set. The validation set is used to tune hyper-parameters and apply early stopping. The hyper-parameters with which the model performs best on the validation set are selected for final evaluation with the test set.

The datasets used in our analyses and ablation studies follow the tiny-benchmark in Ye et al. (2024a), which consists of 45 datasets. The performance rankings of methods on this smaller benchmark are consistent with those on the full benchmark, making it a useful probe for tabular analysis.

### A.3    HARDWARE

The majority of experiments, including those measuring time and memory overhead, were conducted on a Tesla V100 GPU.

### A.4    POTENTIAL ALTERNATIVE IMPLEMENTATION

We explore an alternative strategy to learn the embedding $\phi$ in two steps. First, we apply Supervised Contrastive loss (Sohn, 2016; Khosla et al., 2020), where supervision is generated within a mini-batch. After learning $\phi$, we use KNN for classification or regression during inference. In the regression scenario, label values are discretized, and we refer to this baseline method as Tabular Contrastive (TabCon). Empirically, we find that certain components of MODERNNCA, such as the Soft-NN loss for prediction, cannot be directly applied to TabCon, even when $\phi$ is implemented using the same nonlinear MLP as in MODERNNCA. Despite this, the TabCon baseline remains competitive with FT-Transformer (FT-T), achieving average ranks similar to L-NCA in both classification and regression tasks.

### A.5    COMPARISON METHODS

We compare MODERNNCA with 20 approaches among three different categories. First, we consider **classical parametric methods**, including linear SVM and tree-based methods like RandomForest, XGBoost (Chen & Guestrin, 2016), LightGBM Ke et al. (2017), and CatBoost (Prokhorenkova et al., 2018). Then, we consider **parametric deep models**, including NODE (Popov et al., 2020), MLP (Kadra et al., 2021; Gorishniy et al., 2021), ResNet (Gorishniy et al., 2021), SAINT (Somepalli et al., 2022), DCNv2 (Wang et al., 2021), FT-Transformer (Gorishniy et al., 2021), DANets (Chen et al., 2022), MLP-PLR (Gorishniy et al., 2022), TabCaps (Chen et al., 2023), Tangos (Jeffares et al., 2023), PTaRL (Ye et al., 2024b), SwitchTab (Wu et al., 2024), and ExcelFormer (Chen et al., 2024).

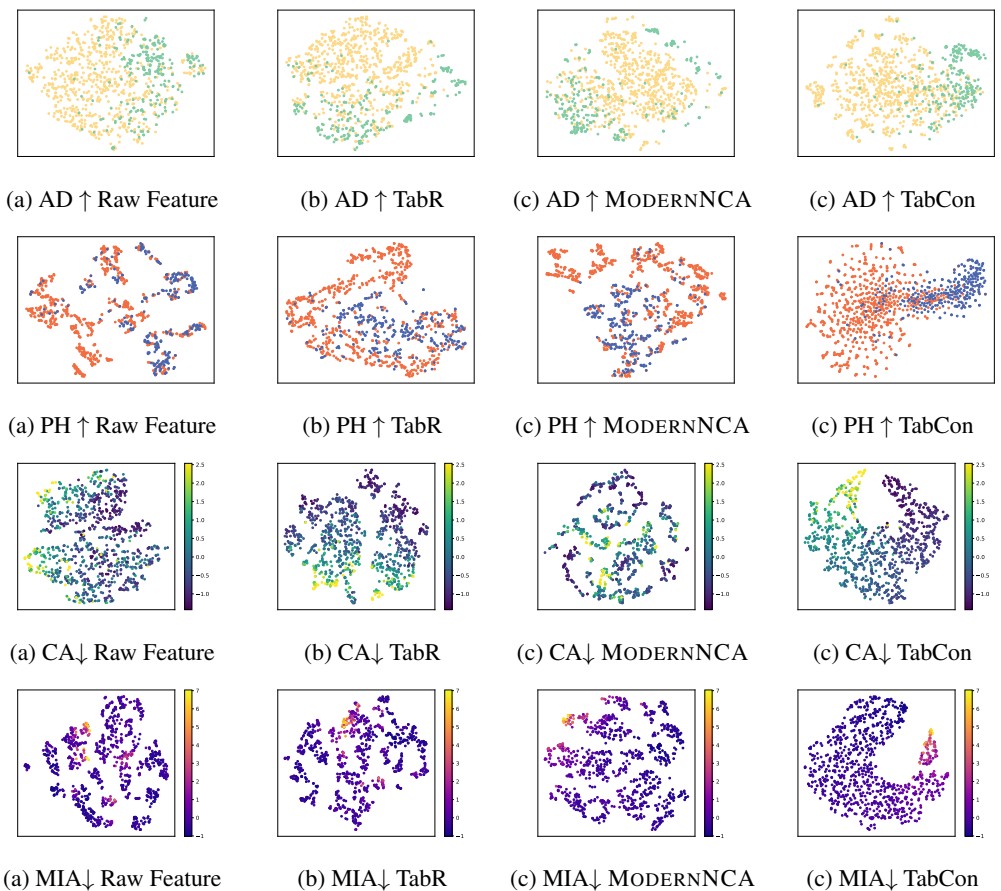

Figure 4: Visualization of the embedding space of different methods.

For **neighborhood-based methods**, we consider KNN and TabR (Gorishniy et al., 2024). For a fair comparison, the same PLR numerical encoding is applied in MLP-PLR, TabR, and MODERNNCA.

## APPENDIX B    ADDITIONAL EXPERIMENTS

### B.1    VISUALIZATION RESULTS

To better analyze the properties of MODERNNCA, we visualize the learned embeddings $\phi(\boldsymbol{x})$ of MODERNNCA, TabCon (mentioned in subsection A.4), and TabR using TSNE (Van der Maaten & Hinton, 2008). As shown in Figure 4, all deep tabular methods transform the embedding spaces to be more helpful for classification or regression compared to the raw features. The embedding space learned by TabCon clusters samples of the same class together and separates samples of different classes, often clustering same-class instances into a single cluster. However, it still struggles with some hard-to-distinguish samples. TabR and MODERNNCA, on the other hand, divide samples of the same class into multiple clusters, ensuring that similar samples are positioned closer to each other. This strategy aligns with the prediction mechanism of KNN, where good performance is achieved by clustering instances with similar neighbors together rather than into a single cluster. The embedding space learned by MODERNNCA is more discriminative than that learned by TabR. The main reason is that TabR leverages an additional architecture to modify the prediction score for each instance, making the learned embedding space less discriminative compared to MODERNNCA.

Table 5: Comparison among various distances used to implement Equation 4, where Euclid distance is the default choice in MODERNNCA. We show the change in average performance rank (lower is better) across the five configurations on the 45 datasets in the tiny-benchmark.

|                | Euclid | Dot Product | Cosine | Squared Euclid | L1-Norm |
|----------------|--------|-------------|--------|----------------|---------|
| Classification | 2.593  | 3.852       | 2.111  | 2.630          | 3.769   |
| Regression     | 2.500  | 3.222       | 2.529  | 2.722          | 3.889   |

Table 6: Comparison of different loss functions. The log loss used in MODERNNCA, the original NCA's summation loss, the MCML loss, and the t-distribution loss. The change in average performance rank (lower is better) is presented across these four configurations on the 45 datasets in the tiny-benchmark.

|                | MODERNNCA | NCA   | MCML  | t-distribution |
|----------------|-----------|-------|-------|----------------|
| Classification | 2.074     | 2.519 | 3.074 | 2.333          |
| Regression     | 1.500     | -     | -     | 1.540          |

## B.2 ADDITIONAL ABLATION STUDIES

**The Influence of Distance Functions**. The predicted label of a target instance $\boldsymbol{x}_i$ is determined by the label of its neighbors in the learned embedding space projected by $\phi$. The distance function $\mathrm{dist}(\cdot, \cdot)$ is the key to determining the pairwise relationship between instances in the embedding space and influences the weights in Equation 4.

In MODERNNCA, we choose Euclidean distance

$$\mathrm{dist}_{\mathrm{EUC}}(\phi(\boldsymbol{x}_i), \phi(\boldsymbol{x}_j)) = \sqrt{(\phi(\boldsymbol{x}_i) - \phi(\boldsymbol{x}_j))^{\top}(\phi(\boldsymbol{x}_i) - \phi(\boldsymbol{x}_j))} = \|\phi(\boldsymbol{x}_i) - \phi(\boldsymbol{x}_j)\|_2 . \quad (6)$$

We also utilize other distance functions, *e.g.*, the squared Euclidean distance, $\mathrm{dist}_{\mathrm{EUC}}^2(\phi(\boldsymbol{x}_i), \phi(\boldsymbol{x}_j))$, the $\ell_1$-norm distance

$$\mathrm{dist}(\phi(\boldsymbol{x}_i), \phi(\boldsymbol{x}_j)) = \|\phi(\boldsymbol{x}_i) - \phi(\boldsymbol{x}_j)\|_1 , \quad (7)$$

the (negative) cosine similarity $\mathrm{dist}(\phi(\boldsymbol{x}_i), \phi(\boldsymbol{x}_j)) = -(\boldsymbol{x}_i^{\top}\boldsymbol{x}_j)/(\|\boldsymbol{x}_i\|_2\|\boldsymbol{x}_j\|_2)$, and the (negative) inner product $\mathrm{dist}(\phi(\boldsymbol{x}_i), \phi(\boldsymbol{x}_j)) = -\phi(\boldsymbol{x}_i)^{\top}\phi(\boldsymbol{x}_j)$. The results using different distance functions are listed in Table 5, which contains the average performance rank over 45 datasets among the five variants. On average, Euclidean distance performs well across both classification and regression tasks. While cosine distance yields better results on classification datasets (with an average performance rank of 4.5939 compared to MODERNNCA and 20 other methods across 300 datasets, please check Figure 2 for details), its advantage diminishes on regression tasks.

**Other Possible Loss Functions**. NCA (Goldberger et al., 2004) originally explored two loss functions: one that maximizes the sum of probabilities in Equation 3, and another that minimizes the sum of log probabilities as in Equation 1. The former was selected in the original implementation of NCA due to its better performance. We also investigated several alternative loss functions for NCA. For instance, MCML (Globerson & Roweis, 2005) minimizes the KL-divergence between the learned embedding in Equation 2 and a constructed ground-truth label distribution for each instance, but it only applies to classification tasks. Another variant is the t-distributed NCA (Min et al., 2010), which uses a heavy-tailed t-distribution to measure pairwise similarities in the objective function. We tested both MCML and the t-distribution loss functions in MODERNNCA, and the results are summarized in Table 6, showing the average ranks across 45 datasets. The log objective in Equation 1 performs best for classification tasks and slightly outperforms the t-distribution variant in regression tasks.

**The Influence of Sampling Strategy**. As mentioned before, SNS randomly samples a subset of training data for each mini-batch when calculating the loss of Equation 4. We also investigate whether we could further improve the classification/regression ability of the model when we incorporate richer information during the sampling process, *e.g.*, the label of the instances.

We consider two other sampling strategies in addition to the fully random one we used before. First is class-wise random sampling, which means that given a proportion, we sample from each class

Table 7: Comparison of different sampling strategies: "Random", "Label", and "Distance" represent MODERNNCA's naive uniform sampling, class-wise random sampling, and distance-based sampling, respectively. The change in average performance rank (lower is better) is presented across these three configurations on the 45 datasets in the tiny-benchmark.

|                | Random | Label | Distance |
|----------------|--------|-------|----------|
| Classification | 1.869  | 2.230 | 1.901    |
| Regression     | 1.508  | -     | 1.492    |

Table 8: Comparison of various architecture choices based on a fixed 2-layer MLP. We only tune architecture-independent hyper-parameters for different variants. The change in average performance rank (lower is better) is shown across three configurations (default, Layer Norm, and Residual) on the 45 datasets in the tiny-benchmark.

|                | MLP   | w/ LayerNorm | ResNet |
|----------------|-------|--------------|--------|
| Classification | 1.905 | 2.048        | 2.048  |
| Regression     | 1.813 | 2.313        | 1.875  |

in the training set and combine them together. This strategy takes advantage of the training label information and keeps the instances from all classes that will exist in the sampled subset. Besides, we also consider the sampling strategy based on the pairwise distances between instances. Since the neighbors of an instance may contribute more (with larger weights) in Equation 4, so given a mini-batch, we first calculate the Euclidean distance between instances in the batch and all the training set with the embedding function $\phi$ in the current epoch. Then we sample the training set based on the reciprocal of the pairwise distance value. In detail, given an instance $x_i$, we provide instance-specific neighborhood candidates and $x_j$ in the training set is sampled based on the probability $\sim 1/(\text{dist}(\phi(x_i), \phi(x_j)))^\tau$. $\tau$ is a non-negative hyper-parameter to calibrate the distribution. The distance calculation requires forward passes of the model $\phi$ over all the training instances, and the instance-specific neighborhood makes the loss related to a wide range of the training data. Therefore, the distance-based sampling strategy has a low training speed and high computational burden.

The comparison results, *i.e.*, the average performance rank, among different sampling strategies on 45 datasets are listed in Table 7. We empirically find the label-based sampling strategy cannot provide further improvements. Although the distance-based strategy helps in certain cases, the improvements are limited. Taking a holistic consideration of the performance and efficiency, we choose to use the vanilla random sampling in MODERNNCA.

**Comparison between Different Deep Architectures**. Unlike the ablation studies in subsection 6.2, where we fixed the model family and tuned detailed hyper-parameters (such as the number of layers and network width) based on the validation set, here we fix the main architecture as a two-layer MLP and only tune architecture-independent hyper-parameters, such as the learning rate.

With this base MLP architecture, we evaluate three variants: the base MLP, one with batch normalization replaced by layer normalization, and one with an added residual link. The average ranks of the three variants across 45 datasets are presented in Table 8. We observe that the basic MLP remains a better choice compared to the versions with a residual link or layer normalization.

### B.3   RUN-TIME AND MEMORY USAGE ESTIMATION

We make a run-time and memory usage comparison in Figure 1. Here are the steps that we take to perform the estimation. First, we tuned all models on the validation set for 100 iterations, saving the optimal parameters ever found. Next, we ran the models for 15 iterations with the tuned parameters and saved the best checkpoint on the validation set. The run-time for the models was estimated using the average time taken by the tuned model to run one seed in the training and validation stage.

We present the average results of run-time and memory usage estimation across the full benchmark (300 datasets) in Table 9.

Table 9: Training time and memory usage estimation for different tuned models over 300 datasets. The average rank represents the mean performance ranking of these models based on the performance metrics (RMSE for regression and accuracy for classification).

| Model | M-NCA | L-NCA | MLP | MLP-PLR | FT-T | TabR | XGBoost | CatBoost |
|---|---|---|---|---|---|---|---|---|
| Training Time (s) | 87.5 | 33.62 | 30.36 | 42.87 | 111.91 | 173.34 | 4.53 | 20.48 |
| Memory Usage (GB) | 5.36 | 1.42 | 1.15 | 2.37 | 4.98 | 10.13 | 0.84 | 1.06 |
| Average Rank | 4.56 | 6.30 | 7.53 | 6.94 | 6.29 | 5.36 | 5.62 | 4.61 |

## B.4    FULL RESULTS ON THE BENCHMARK

Due to the extensive size of the results table, we have uploaded the complete performance metrics of ModernNCA alongside other comparison methods, including RealMLP (Holzmüller et al., 2024), at `https://github.com/LAMDA-Tabular/TALENT/tree/main/results`.

## B.5    LIMITATIONS

ModernNCA has two possible limitations.

The first limitation pertains to handling tabular data with distribution shifts, as discussed in Rubachev et al. (2025). Specifically, ModernNCA does not explicitly account for implicit temporal relationships between instances and their neighbors during the neighborhood search. However, a recent study (Cai & Ye, 2025) has shown that adopting alternative data-splitting protocols—such as random splits for training and validation—significantly improves ModernNCA's performance, making it competitive with other methods. Furthermore, ModernNCA's performance is further enhanced when incorporating temporal embeddings.

The second limitation lies in handling high-dimensional datasets where $d \gg N$ (Jiang et al., 2024), as observed in Ye et al. (2025). This challenge is well-known in classical metric learning (Shi et al., 2014; Liu et al., 2015), where distance calculations become less reliable due to the curse of dimensionality. High-dimensional data can lead to reduced neighborhood retrieval effectiveness, impacting prediction accuracy. Potential mitigations include pre-processing with dimensionality reduction techniques and leveraging ensemble approaches (Liu & Ye, 2025), which may help alleviate the adverse effects of high dimensionality.

