# OpenReview forum: "Revisiting Nearest Neighbor for Tabular Data: A Deep Tabular Baseline Two Decades Later"
_ICLR.cc/2025/Conference — ICLR 2025 Poster_

### Official Review · Reviewer_B7pB · 2024-10-28

**Soundness:** 3
**Presentation:** 3
**Contribution:** 3
**Rating:** 8
**Confidence:** 4

**Summary:**

This paper re-evaluates the nearest neighbor approach for tabular data, proposing a modernized version of neighborhood component analysis (nca) called modernnca. starting from classical nn techniques, the authors incorporate current deep learning methods into nca, such as using stochastic gradient descent and adding stochastic neighborhood sampling. the experiments on 300 tabular datasets demonstrate that modernnca performs comparably to leading models like catboost, outperforming other deep learning approaches for classification and regression tasks. the authors also provide insights into how modern techniques like batch normalization and non-linear architectures improve nn performance.

**Strengths:**

1. the authors structure their explanation to make complex methods like nn and nca accessible and coherent. this clarity is helpful for understanding both the motivation and methodology behind the modifications to nca.

2. the paper includes an extensive evaluation on a broad range of datasets, demonstrating modernnca’s effectiveness with detailed performance metrics and statistical significance tests.


3.  The authors do a great job explaining the steps taken to modernize nca

**Weaknesses:**

W1: while the model shows high performance, the paper lacks specific scenarios or guidance on applying modernnca practically, such as when dealing with imbalanced datasets

**Questions:**

1. How does modernnca perform with different distance metrics, such as cosine similarity, especially for datasets with high-dimensional features?


2. Given that modernnca outperformed many models, have you tested its performance on imbalanced datasets?

3. Could you provide more insights into cases where modernnca significantly underperforms compared to catboost or other tree-based methods?

---

> ### Author Response · Authors · 2024-11-21
> **Response**
>
> Thanks for the valuable feedback. In this rebuttal, we address the reviewers' suggestions and concerns in a Q&A format:
>
> **Q1**:How does ModernNCA perform with different distance metrics, such as cosine similarity, especially for datasets with high-dimensional features?
> **A1**: Please refer to A4 for reviewer 3moZ.
>
> **Q2:**  while the model shows high performance, the paper lacks specific scenarios or guidance on applying modernnca practically, such as when dealing with imbalanced datasets. Given that ModernNCA outperformed many models, have you tested its performance on **imbalanced** datasets?
>
>
> **A2:** Thank you for this insightful question. To investigate the performance of ModernNCA on imbalanced datasets, we selected 64 datasets from the 180 classification datasets used in our main experiments. These datasets were identified as imbalanced based on a class ratio greater than 5 between the major and minor classes. In this case, we used F1 as the evaluation metric, as it accounts for class imbalance.
>
> The average performance ranks of representative methods are listed below (lower rank indicates better performance):
>
> | **Method**       | **Rank** |
> |-------------------|----------|
> | ModernNCA (post-weight)    |3.54     |
> | ModernNCA        | 4.41     |
> | CatBoost         |4.25     |
> | XGBoost          | 3.95     |
> | RandomForest     | 6.19     |
> | TabR             | 4.68     |
> | MLP              | 6.14     |
> | MLP-PLR          |5.34     |
> | FT-Transformer   | 5.25     |
>
>
> We have the following two observations:
>
> 1. **Competitive Performance:** ModernNCA achieves a rank comparable to the strongest method, CatBoost, and outperforms other deep tabular methods, including TabR and MLP variants.
> 2. **TabR Limitations:** TabR struggles on imbalanced datasets compared to more balanced cases. A likely reason is that the imbalanced label distribution makes it difficult for TabR to retrieve meaningful neighbors effectively.
>
> Therefore, ModernNCA demonstrates robustness on imbalanced datasets due to its reliance on a learned embedding space that clusters similar instances effectively, regardless of class distribution. This clustering property ensures that neighbors belonging to minority classes are still well-represented in the embedding space.
>
> Moreover, ModernNCA's simplicity makes it amenable to standard techniques for handling imbalance. For example: we can adjust the weights of Neighbors with a learned ModernNCA model, where The influence of neighbors can be adjusted by assigning higher weights to minority class instances. We denote this version as ModernNCA (post-weight), which achieves the best performance among all other models.
>
> These observations and practical suggestions will be included in the final version of the paper to provide more actionable insights for practitioners.
>
>
> **Q3**: Could you provide more insights into cases where ModernNCA significantly underperforms compared to catboost or other tree-based methods?
>
> **A3**: Please refer to A3 for reviewer Cjck.

---

> > ### Comment · Reviewer_B7pB · 2024-11-25
> >
> > Thanks - I have no other concerns.

---

### Official Review · Reviewer_3moZ · 2024-11-01

**Soundness:** 2
**Presentation:** 3
**Contribution:** 2
**Rating:** 5
**Confidence:** 4

**Summary:**

This paper explores the potential of modernizing the classical Nearest Neighbor approach for tabular data by leveraging a differentiable K-nearest neighbors variant, Neighborhood Components Analysis (NCA). The authors introduce MODERNNCA, an improved version of NCA that integrates deep learning techniques such as stochastic gradient descent (SGD), nonlinear embeddings, and a Stochastic Neighborhood Sampling (SNS) strategy to boost computational efficiency and model performance. They demonstrate that MODERNNCA matches or outperforms both tree-based models and current deep tabular models across 300 datasets in classification and regression tasks.

**Strengths:**

**Originality**
Revisiting a classic nearest-neighbor approach with contemporary deep learning techniques is a novel approach, particularly since NCA had been previously limited by computational efficiency and scalability. This approach aims to unify insights from both traditional and modern tabular prediction methods.

**Quality**
The authors conducted extensive experiments across 300 datasets, providing thorough evidence of the model’s strengths and weaknesses.

**Clarity**
The explanation of the modifications, including SGD, nonlinear embeddings, and SNS, is detailed and clear, making the improvements accessible to the reader.

**Weaknesses:**

**Marginal Contribution**
   The paper’s contribution feels incremental rather than pioneering. The improvements in MODERNNCA rely on established techniques (SGD, SNS, and nonlinear embeddings) without introducing a fundamentally new concept or method. This makes the novelty limited, as it essentially optimizes an existing algorithm rather than providing a unique advancement.

**Limited Novelty in Comparison to KNN Variants**
   The paper lacks direct comparisons with other KNN-inspired deep learning methods that have similarly benefited from modern optimization strategies. This limited scope of comparison weakens the argument for MODERNNCA’s distinctiveness and impact.

**Lack of Theoretical Analysis**
   The paper’s focus is primarily empirical, with little theoretical exploration of why the proposed enhancements lead to improved performance. A deeper theoretical perspective on the modifications—such as the effect of stochastic sampling on generalization—would provide valuable insights and strengthen the work's academic contribution.

**Questions:**

- MODERNNCA relies on Euclidean distance as the default metric. How adaptable is the model to other distance metrics, such as cosine similarity, and do the authors have insights on how these choices might impact performance?

- The paper discusses various deep learning modifications, but how sensitive is MODERNNCA to hyperparameter choices such as learning rate, number of neighbors (K), and embedding dimensions? Are there specific configurations where performance significantly degrades?

---

> ### Author Response · Authors · 2024-11-21
> **Response (1/4)**
>
> Thanks for the valuable feedback. In this rebuttal, we address the reviewers' suggestions and concerns in a Q&A format:
>
> **Q1**: Marginal Contribution. The paper’s contribution feels incremental rather than pioneering. The improvements in MODERNNCA rely on established techniques (SGD, SNS, and nonlinear embeddings) without introducing a fundamentally new concept or method. This makes the novelty limited, as it essentially optimizes an existing algorithm rather than providing a unique advancement.
>
>
> **A1:** Thanks. We would like to clarify the value and context of ModernNCA’s contributions, emphasizing that while the core techniques used in the paper are possibly established, their effective integration to achieve strong performance on tabular tasks represents a significant advancement. We have provided a detailed response to a similar concern raised by review Qykr. In the following, we further highlight our technical contributions and significance.
>
> 1. **The Value of Making Existing Ideas Work Effectively**. While ModernNCA builds upon existing components like SGD and nonlinear embeddings, the paper demonstrates how to combine these elements to optimize the performance of Neighborhood Component Analysis (NCA) for tabular data. This approach bridges a gap between classical metric learning and modern deep learning, which, to the best of our knowledge, has not been successfully achieved before.
> 2. **The mindset behind ModernNCA is grounded in solving real-world problems by leveraging what we already know and enhancing it through thoughtful integration and design**. This philosophy aligns with many impactful works in machine learning, where the emphasis is not solely on theoretical novelty but also on advancing the utility and practicality of established methods. For example, the adoption of SGD for training NCA might seem incremental, but it significantly improves scalability and generalization in real-world scenarios; SNS improves efficiency while preserving or even enhancing performance, a critical requirement for handling large-scale datasets.
> 3. **Revisiting Classical Methods for Modern Applications**. ModernNCA is not merely an incremental optimization; it represents a paradigm shift in how classical ideas can be revisited and adapted for contemporary challenges. Our observations show that simple adjustments like switching to a soft nearest neighbor loss, introducing nonlinear architectures, and employing SNS unlock substantial performance gains, surpassing state-of-the-art methods in many cases; the fact that ModernNCA can consistently compete with and often outperform sophisticated models like CatBoost and TabR highlights the practical significance of these contributions.
>
> We will incorporate these clarifications into the final version of the paper to better articulate the significance of ModernNCA's contributions.

---

> ### Author Response · Authors · 2024-11-21
> **Response (2/4)**
>
> **Q2: Limited Novelty in Comparison to KNN Variants. The paper lacks direct comparisons with other KNN-inspired deep learning methods that have similarly benefited from modern optimization strategies. This limited scope of comparison weakens the argument for MODERNNCA’s distinctiveness and impact.**
>
>
> **A2:** Thank you for the comment. We would like to address the concerns regarding novelty and distinctiveness by clarifying the comparisons included in the paper and adding further context. First of all, we have included comparisons with two representative KNN-inspired deep-learning methods (TabR and TabCon) in our submission.
>
> 1. **Existing Comparisons with KNN Variants.** In this paper, we directly compare ModernNCA with multiple KNN-based methods, including:
>    * **Vanilla KNN:** A baseline non-parametric approach using raw feature distances.
>    * **TabR:** A strong neighborhood-based deep tabular model.
>    * **TabCon:** Our proposed baseline model, which utilizes supervised contrastive loss during training as ModernNCA and applies KNN over the learned embeddings in inference (described in Appendix A.4).
> 2. These comparisons highlight the improvements ModernNCA achieves by combining classical ideas with modern deep learning techniques.
>
> To strengthen the argument for ModernNCA’s distinctiveness, we extend our comparisons to include PFN-KNN \[A\], a recently proposed KNN-based method *specifically designed for classification datasets*. We evaluate these methods on 45 datasets from our ablation studies. The results, shown below, indicate that ModernNCA maintains its superiority over other KNN variants:
>
> | **Method**   | **Classification (cls)** | **Regression (reg)** | **Overall (all)** |
> |--------------|---------------------------|-----------------------|-------------------|
> | CatBoost     | 3.777778                 | 2.777778             | 3.377778         |
> | ModernNCA    | 3.666667                 | 3.500000             | 3.600000         |
> | TabR         | 3.814815                 | 3.647059             | 3.750000         |
> | XGBoost      | 4.037037                 | 3.764706             | 3.931818         |
> | MLP-PLR      | 5.370370                 | 4.166667             | 4.888889         |
> | PFN-KNN      | 6.222222                 | \-                    | 6.222222         |
> | TabCon       | 6.888889                 | 5.705882             | 6.431818         |
> | KNN          | 7.592593                 | 6.058824             | 7.000000         |
>
> While ModernNCA could incorporate more "fancy" components, we consciously chose to focus on simplicity and practical utility. ModernNCA achieves strong results without unnecessary complexity, making it both effective and accessible.
>
> In summary, ModernNCA establishes itself as a robust KNN-inspired method, leveraging classical principles with modern enhancements. Its superior performance across multiple benchmarks, even against advanced KNN variants like TabR and PFN-KNN, underscores its distinctiveness and practical impact. We will further emphasize these comparisons and their implications in the final version of the paper.
>
> \[A\] Retrieval & Fine-Tuning for In-Context Tabular Models. NeurIPS 2024\.

---

> ### Author Response · Authors · 2024-11-21
> **Response (3/4)**
>
> **Q3**: Lack of Theoretical Analysis. The paper’s focus is primarily empirical, with little theoretical exploration of why the proposed enhancements lead to improved performance. A deeper theoretical perspective on the modifications would provide valuable insights and strengthen the work's academic contribution.
>
> **A3:** Thank you for this suggestion. We acknowledge that the inclusion of a theoretical analysis would strengthen our work, and it applies to all deep tabular methods as well. However, conducting theoretical analyses is non-trivial, especially for deep methods. Often, one needs to make strong assumptions, which inevitably degrade the applicability of the analysis. To our knowledge, very few deep tabular methods have included theoretical analyses.
>
> That said, nearest-neighbor-based methods have a solid theoretical foundation. Our modifications, though without formal theoretical analysis, were well-motivated to address the challenges we observed. We deliberately kept our modifications simple and intuitive and validated them with extensive experiments. Put together, we respectfully think that our empirical evidence offers valuable insights for future work to build upon.
>
> **Empirical Strength and Simplicity.** Our approach aims to build on an established and interpretable method, NCA, by incorporating modern deep learning techniques. These modifications, such as Stochastic Neighborhood Sampling (SNS), are empirically validated across a large-scale tabular benchmark, demonstrating consistent improvements in performance.
>
> **Theoretical Implications:** While we do not delve deeply into theoretical proofs, the proposed enhancements carry intuitive theoretical justifications:
>
> - **Stochastic Sampling and Generalization:** SNS introduces randomness during training, which likely acts as an implicit regularization mechanism, forcing the model to generalize better by learning robust embeddings that are not overly dependent on specific neighbors. This aligns with general theories of stochastic training methods in deep learning.
> - **Soft-NN Loss:** The use of a differentiable soft nearest neighbor loss directly aligns the embedding optimization with the prediction objective, theoretically reducing the mismatch between training and inference objectives compared to hard nearest neighbor methods.
>
> We emphasize that our primary goal was to provide a strong empirical foundation for these techniques, which can serve as a baseline for both practical and theoretical advancements. Our approach’s simplicity is a strength, demonstrating that even straightforward extensions of a classical method can rival or outperform complex models.
>
> In summary, while the theoretical exploration of our enhancements is limited in this paper, we have laid the groundwork for further theoretical studies and welcome future research that builds on our contributions to better formalize their effects.
>
> **Q4: MODERNNCA relies on Euclidean distance as the default metric. How adaptable is the model to other distance metrics, and do the authors have insights on how these choices might impact performance?**
>
> **A4:** Thank you for raising this question. We evaluated the adaptability of ModernNCA to alternative distance metrics, including cosine distance, L1-distance, dot product, and squared L2-distance. The results of these experiments are provided in **Appendix B.2** (**Table 5**). Below, we summarize our key observations:
>
> 1. **Performance Overview:** On average, Euclidean (L2) distance performs well across both classification and regression tasks, making it a robust default choice.
>    * **Cosine Distance** demonstrates superior performance on classification datasets (achieving an average performance rank of 4.5939 when compared with ModernNCA and 20 other methods across 300 datasets). However, its advantage diminishes on regression tasks, likely due to its focus on angular relationships rather than magnitude.
>    * **L1-Distance** performs comparably to L2 on some tasks but has higher computational costs due to the absolute operations involved.
>    * **Squared L2-Distance** slightly underperforms compared to standard L2, likely because the additional squaring step impacts gradient behavior during optimization.
> 2. **Why we use L2 as the default choice:**
>     * **Standard Practice:** L2 has long been a standard metric in machine learning and deep learning, particularly in early methods like k-NN and NCA. Its simplicity, isotropic scaling, and compatibility with Euclidean geometry make it a reliable choice across diverse tasks.
>    * **Gradient Behavior:** In optimization, L2 provides smooth and consistent gradients, which can enhance training stability. Metrics like L1 can lead to discontinuous gradients, making optimization more challenging.
>    * **Adaptability:** Euclidean distance effectively balances both magnitude and directional information, which is particularly useful in high-dimensional embedding spaces.

---

> > ### Author Response · Authors · 2024-11-21
> > **Response (4/4)**
> >
> > **Q5: The paper discusses various deep learning modifications, but how sensitive is ModernNCA to hyperparameter choices such as learning rate, number of neighbors (K), and embedding dimensions? Are there specific configurations where performance significantly degrades?**
> >
> >
> > **A5:** Thank you for the question. Due to the high-dimensional nature of tabular data and the diversity of datasets, it is reasonable to expect that different datasets need different hyperparameter configurations. For instance, datasets with more complex distributions demand higher embedding dimensions, but such a choice also depends on whether we have sufficient training samples. To address this, we follow recent studies and tune hyperparameters using the toolbox Optuna, which enables efficient and automatic hyperparameter search. Moreover, since ModernNCA adopts the soft-NN strategy in both the training and the evaluation stages, all the neighbors are incorporated in a weighted manner to obtain the final classification or regression results. In other words, we do not need to determine the value of K in ModernNCA.
> >
> > We provide results to evaluate the sensitivity of ModernNCA to two key hyperparameters: learning rate and embedding dimensions. For each experiment, other hyperparameters were set to their default values, and tests were conducted over 45 datasets as described in Section 6\.
> >
> > | **Learning Rate** |**Classification** | **Regression** | **All**    |
> > |--------------------|--------------------|----------------|------------|
> > | ModernNCA (tuned) | 1.70               | 1.78           | 1.73       |
> > | LR-0.1            | 4.37               | 4.83           | 4.56       |
> > | LR-0.01           | 2.44               | 2.89           | 2.62       |
> > | LR-0.001          | 2.63               | 2.56           | 2.60       |
> > | LR-0.0001         | 3.70               | 2.94           | 3.40       |
> >
> > **Observation**:Learning rates in the range of **0.01–0.001** lead to better results than other configurations.
> >
> > We tested ModernNCA with different embedding dimensions. The average ranks for classification, regression, and all datasets are listed below. In ModernNCA, we tune the dimensions for different datasets over their training set.
> >
> > | **Dimension**     | **Classification** | **Regression** | **All**    |
> > |--------------------|--------------------|----------------|------------|
> > | ModernNCA (tuned dimensions) | 2.37               | 2.50           | 2.42       |
> > | Dim-10            | 5.11               | 5.39           | 5.22       |
> > | Dim-20            | 4.74               | 5.06           | 4.87       |
> > | Dim-50            | 3.78               | 4.28           | 3.98       |
> > | Dim-100           | 4.96               | 3.89           | 4.53       |
> > | Dim-200           | 3.07               | 3.67           | 3.31       |
> > | Dim-500           | 3.15               | 3.22           | 3.18       |
> >
> >
> > **Observation**: Larger embedding dimensions generally result in better performance, with dimensions around **200–500** providing a good balance of model capacity and performance.
> >
> > In summary, ModernNCA shows strong robustness with tuned learning rates and embedding dimensions. Learning rates between **0.01–0.001** and embedding dimensions in the range of **200–500** tend to yield the best results.

---

> > > ### Comment · Reviewer_3moZ · 2024-11-25
> > >
> > > Thank you for the response. While you demonstrate comparisons with various KNN-based methods (Vanilla KNN, TabR, and TabCon) and explain ModernNCA's combination of classical and modern techniques, the core limitation of methodological novelty remains a significant concern. The primary contribution still appears to be mainly an application of existing techniques to a new context, rather than presenting fundamental innovations. Therefore, I maintain my current score.

---

> > > > ### Author Response · Authors · 2024-12-04
> > > >
> > > > Dear reviewer Reviewer 3moZ,
> > > >
> > > > Thank you for your additional comment. We provide further responses as follows.
> > > >
> > > > Q1: While you demonstrate comparisons with various KNN-based methods (Vanilla KNN, TabR, and TabCon) and explain ModernNCA's combination of classical and modern techniques, the core limitation of methodological novelty remains a significant concern.
> > > >
> > > > A1: We appreciate the reviewer’s recognition of the comparisons between ModernNCA and various KNN-based methods (Vanilla KNN, TabR, and TabCon), where ModernNCA demonstrates superior performance across several metrics.
> > > > *In terms of novelty, we respectfully think its definition is not limited to creating something new, such as new model architectures or objective functions. In our humble opinion, demonstrating that “a previously believed inferior approach can indeed achieve decent performance” is novel and significant, and we view it as our main contribution.*
> > > >
> > > > While many existing deep tabular methods focused on developing new models or training algorithms, we think one fundamental mindset in machine learning — the principle of Occam’s Razor — should not be abandoned. In our humble opinion, if one can revitalize a traditional approach to be on par with leading methods, it should be encouraged (as a strength) rather than discouraged (as a weakness), especially because many prior attempts have failed.
> > > >
> > > > For full disclosure, in pursuing this project, our first try was to extend and improve TabR. However, after a closer investigation, we found the key to success is the use of nearest neighbors. This motivated us to take an alternative route, starting from the basic nearest-neighbor method and seeking to step-by-step turn it into a strong deep tabular baseline while knowing that taking such a route might adversely hurt our “novelty.” That said, we still believe as a research community of machine learning, the principle of Occam’s Razor and the investigation into traditional methods should be encouraged.
> > > >
> > > > Q2: The primary contribution still appears to be mainly an application of existing techniques to a new context, rather than presenting fundamental innovations.
> > > >
> > > > A2: We would like to note that nearest-neighbor methods such as NCA were developed in the prior deep learning era when each data instance was assumed to be a pre-defined feature vector. In other words, *they were originally developed for tabular data* but were found much inferior compared to leading tree-based methods like CatBoost, XGBoost, and LightGBM.
> > > >
> > > > Therefore, in our humble opinion, *our contribution is NOT “applying existing techniques to a new context” BUT “modernizing them for the context they should have thrived.”* We systematically revisit NCA and revitalize it to excel in tabular data. We want to reiterate that doing so is never easy. Often, it is much more difficult than developing a new method. We thus humbly view our work as a fundamental contribution to machine learning, rather than a superficial direct application. It has a profound implication that an appropriate integration of traditional methods and deep learning techniques could lead to strong approaches to tabular data.

---

### Official Review · Reviewer_Qykr · 2024-11-01

**Soundness:** 2
**Presentation:** 2
**Contribution:** 1
**Rating:** 3
**Confidence:** 4

**Summary:**

The paper revisits the Neighborhood Components Analysis (NCA) and adapts it for tabular data learning, proposing ModernNCA as an enhanced approach. The modifications include (1) calculating distances in a representation space, (2) using stochastic gradient descent (SGD) instead of L-BFGS for optimization, and (3) training in a mini-batch fashion rather than on the entire dataset at once. The authors benchmarked ModernNCA against numerous methods across 300 datasets, finding that it achieved consistently superior performance, often comparable to leading models like CatBoost and outperforming many deep tabular learning methods.

**Strengths:**

- The approach effectively leverages modern deep learning techniques to enhance classical NCA, demonstrating strong empirical results across a large number of datasets.

- The paper provides comprehensive benchmarks, including comparisons with state-of-the-art methods in both classification and regression tasks.

**Weaknesses:**

**Lack of Novelty**: While the paper shows strong empirical performance, the core modifications (using a representation space for distance calculations, employing SGD, and mini-batch training) have already been explored in prior research. This raises concerns regarding the originality of the contribution. The changes appear more like tunings of established techniques rather than introducing a fundamentally new method.

[Prior research example] J Kang et al., Deep metric learning based on scalable neighborhood components for remote sensing scene characterization, 2020.

**Questions:**

If my understanding is incorrect, could you please clarify what is the novel concept introduced in this paper?

---

> ### Author Response · Authors · 2024-11-21
> **Response (1/2)**
>
> Thanks for the valuable feedback. In this rebuttal, we address the reviewers' suggestions and concerns in a Q&A format:
>
> **Q**: Lack of Novelty: While the paper shows strong empirical performance, the core modifications (using a representation space for distance calculations, employing SGD, and mini-batch training) have already been explored in prior research. This raises concerns regarding the originality of the contribution. The changes appear more like tunings of established techniques rather than introducing a fundamentally new method.
>
> [Prior research example] J Kang et al., Deep metric learning based on scalable neighborhood components for remote sensing scene characterization, 2020.
>
> If my understanding is incorrect, could you please clarify what is the novel concept introduced in this paper?
>
>
> **A:** Thanks. We appreciate the opportunity to clarify what sets our work apart.
>
> 1. **Reviving NCA for Tabular Data.** While metric learning and neighborhood-based methods have seen extensive exploration in fields like computer vision (e.g., image retrieval) and remote sensing, their usage in tabular data, where they were originally designed, has been limited due to their less competitive performance. This remains the case even with the incorporation of deep learning techniques (please kindly see the remark in section 1 of the main paper). Therefore, we respectfully think how to **effectively** turn a classical method like NCA into a competitive tabular baseline itself is a significant and impactful contribution, regardless of whether each piece of our modifications has been explored somewhere in the literature.
>    As an analogy, diffusion models have been brought into machine learning and computer vision since 2016; contrastive learning objectives have been well-known since early 2000\. However, how to transform them into state-of-the-art generative models and self-supervised learning objectives is non-trivial, often involving insights into the detailed implementation. Likewise, NCA has existed for decades, but its potential for tabular data has not been fully realized. Our work demonstrates the importance of a thorough and thoughtful investigation in bridging the gap between theory and practical application.
>
> 2. Specifically, **many prior works**, including the mentioned paper by J. Kang et al. (2020), focus on vision-specific tasks. Their adaptations of NCA often incorporate metric learning losses (e.g., triplet loss) to fine-tune representations for image-based tasks. However, these methods do not address the specific challenges of **tabular data**, where feature sparsity, distribution skewness, and feature-engineering nuances differ significantly from vision domains. In contrast, **our contribution** is a focused rethinking of NCA tailored for tabular data, where design choices like Stochastic Neighborhood Sampling (SNS) and soft nearest neighbor loss in the inference stage play a pivotal role in addressing the scale and complexity of tabular datasets.
> 3. **The Value of Simplicity.** Simple methods that consistently work well across diverse datasets are crucial for practical machine learning. The performance of ModernNCA across 300 datasets, including classification and regression tasks, shows that simplicity need not sacrifice effectiveness.
> 4. **Novel Insights for Tabular Data:**
>    We introduced several key ideas that are novel in the context of tabular data:
>    * **Stochastic Neighborhood Sampling (SNS):** A principled and efficient approach for scaling neighborhood-based methods to large datasets, significantly reducing computational overhead while improving generalization.
>    * **Soft Nearest Neighbor Loss in Prediction:** By aligning the loss function with the prediction strategy, we improve the robustness and scalability of neighborhood-based methods.
>    * **Deep Architectures for NCA:** The paper explores how nonlinear representations and modern optimization strategies (e.g., SGD) fundamentally change the dynamics of NCA for tabular data, enabling it to outperform even tree-based methods like CatBoost.
>
>
> 5.  **Why This Work Matters:** Metric learning is indeed a well-trodden path in vision research, but its adaptation to tabular data has been limited, particularly in creating methods that are efficient, scalable, and effective across diverse tabular datasets. ModernNCA:
>    * Outperforms contemporary deep tabular models like TabR and FT-Transformer on 300 datasets, including challenging real-world scenarios.
>    * Demonstrates how simple yet carefully integrated changes can transform a classical method into a state-of-the-art approach for a new domain.

---

> ### Author Response · Authors · 2024-11-21
> **Response (2/2)**
>
> In sum, while the techniques we employ (e.g., SGD, representation space learning) are not novel in isolation, their **specific adaptation to tabular data** and the resulting **empirical performance** represent a significant contribution. We position ModernNCA as a practical and competitive baseline, which, like foundational works in other domains (e.g., SimCLR, diffusion models), lays the foundation for future exploration and development.
>
> We hope this clarifies the novelty and relevance of our work. We will incorporate these explanations into the final version of the paper for clarity.

---

> ### Comment · Reviewer_Qykr · 2024-11-25
>
> Thank you for your response. As acknowledged by the authors, the primary novelty of this paper lies in applying the well-established NCA, a technique commonly used in other domains, to the tabular domain to achieve improved performance. However, based on the additional insights provided by the authors and results discussed in related works such as TabReD, the proposed approach does not demonstrate clear superiority over other DNN-based methods or GBDTs in terms of performance. This raises concerns that the reported results may reflect thorough tuning on specific datasets rather than a generalizable improvement.
>
> Even without referencing external works, the results reported in this paper do not show significant superiority over other methods. Given that the methodology is not novel and the performance improvements are marginal, the primary contribution of this paper is limited to the application of NCA to the tabular domain. Additionally, the paper does not provide substantial insights or theoretical advancements in the discussion section, further limiting its impact.
>
> In its current form, this paper lacks the innovation and contribution expected of ICLR-quality work. Therefore, I maintain my current score.

---

> ### Author Response · Authors · 2024-12-04
> **Further Responses and Clarifications（1/2）**
>
> Dear reviewer Reviewer Qykr,
>
> Thank you for your additional comment. We provide further responses as follows.
>
> Q1: As acknowledged by the authors, the primary novelty of this paper lies in applying the well-established NCA, a technique commonly used in other domains, to the tabular domain to achieve improved performance.
>
> A1: We would like to clarify that the tabular domain is the domain where traditional machine-learning approaches like NCA were originally developed. Before the rise of deep learning, many machine learning methods assumed that the input data was pre-embedded into a fixed-dimensional feature space, aka, tabular data format. What we argued in the rebuttal is that before the application to the other domains like image recognition, nearest-neighbor methods have been widely applied to tabular data, but the performance seems suboptimal as validated in Table 2 (L432-444) in the paper.
>
> Therefore, in our humble opinion, *our contribution is NOT “applying well-established techniques to a new domain” BUT “modernizing them for the domain they should have thrived in.”* We systematically revisit NCA and revitalize it to extend its success from other domains back to the initial tabular domain.
>
> Q2: However, based on the additional insights provided by the authors and results discussed in related works such as TabReD, the proposed approach does not demonstrate clear superiority over other DNN-based methods or GBDTs in terms of performance. …. Even without referencing external works, the results reported in this paper do not show significant superiority over other methods.
>
> A2: We would like to clarify that *our primary contribution is not to claim superiority over existing leading methods, but to demonstrate a previously believed inferior approach can indeed achieve decent, competitive performance in the tabular domain.* We respectfully view such a contribution as significant, as it opens up new directions for future research in tabular data. It has a profound implication that an appropriate integration of traditional methods and deep learning techniques could lead to strong approaches to tabular data. (We also humbly think that achieving SOTA accuracy is not a requirement for paper acceptance.)
>
> Regarding the discussion in TabReD, ModernNCA was mentioned as impressive on standard classification and regression tasks. It was reported as less effective only under distributional changes, which is how TabReD split the training and test data. We note that addressing such shifts and improving out-of-distribution (OOD) generalization is a fundamental and ongoing challenge not just for nearest-neighbor-based methods, but for many machine learning approaches that rely on empirical risk minimization. We plan to explore methods for enhancing neighbor search and robustness under these conditions in future work.
>
> We note that TabReD was put on arXiv on Jun. 27, 2024 (version 1). The discussion of our approach was added in their version 4 on Oct. 24, 2024, after the ICLR 2025 deadline (Oct. 1, 2024). According to the ICLR 2025 Reviewer Guide (https://iclr.cc/Conferences/2025/ReviewerGuide), “We consider papers contemporaneous if they are published within the last four months.” We would also like to reiterate the point made by Reviewer Cjck: "Importantly, I don't think that performance on new benchmarks is a significant issue for the submission, as it still provides a valuable contribution to tabular DL models based on nearest neighbors. Improving performance in new scenarios is worthy of extended future investigations." We agree with this sentiment and will ensure that we acknowledge the limitation of nearest-neighbor-based methods and outline potential avenues for improvements in the final version of the paper.
>
> Q3: This raises concerns that the reported results may reflect thorough tuning on specific datasets rather than a generalizable improvement.
>
> A3: We have evaluated our approach on 300 tabular datasets, with various evaluation criteria such as average rank, critical difference, paired t-test, and shifted geometric mean, which we believe have covered a diverse range of existing tabular problems. We humbly believe such a scale is beyond the scope of “specific datasets.”

---

> > ### Author Response · Authors · 2024-12-04
> > **Further Responses and Clarifications（2/2）**
> >
> > Q4: Given that the methodology is not novel and the performance improvements are marginal, the primary contribution of this paper is limited to the application of NCA to the tabular domain.
> >
> > A4: In terms of novelty, we respectfully think its definition is not limited to creating something new, such as new model architectures or objective functions. In our humble opinion, demonstrating that “a previously believed inferior approach can indeed achieve decent performance” is novel and significant, and we view it as our main contribution.
> >
> > While many existing deep tabular methods focused on developing new models or training algorithms, we think one fundamental mindset in machine learning — the principle of Occam’s Razor — should not be abandoned. In our humble opinion, if one can revitalize a traditional approach to be on par with leading methods, it should be encouraged (as a strength) rather than discouraged (as a weakness), especially because many prior attempts have failed.
> >
> > As such, we humbly request the reviewer to consider the improvement from a comparison not merely to the SOTA, but to the standard application of nearest neighbor methods. As a research community, we believe that exploring different directions to solve a problem should be encouraged. For full disclosure, in pursuing this project, our first try was to extend and improve TabR. However, after a closer investigation, we found the key to success is the use of nearest neighbors. This motivated us to take an alternative route, starting from the basic nearest-neighbor method and seeking to step-by-step turn it into a strong deep tabular baseline while knowing that taking such a route might adversely hurt our “novelty.” That said, we still believe as a research community of machine learning, the principle of Occam’s Razor and the investigation into traditional methods should be encouraged.
> >
> > Q5: Additionally, the paper does not provide substantial insights or theoretical advancements in the discussion section, further limiting its impact. In its current form, this paper lacks the innovation and contribution expected of ICLR-quality work.
> >
> > A5: We apologize if we did not summarize our insights and contributions clearly in the discussion section, and we will improve it.
> > In our humble opinion, theoretical advancements or creating completely new machine-learning approaches are not necessary components of an accepted paper, based on our prior experience with ICLR and as evidenced by many “revisiting” papers. In terms of contributions, we respectfully think “a revitalization of a traditional approach to making it on par with leading approaches in tabular data” is significant and has profound implications, and our systematic investigation has brought up insights into how such a revitalization is possible.

---

### Official Review · Reviewer_Cjck · 2024-11-01

**Soundness:** 3
**Presentation:** 3
**Contribution:** 3
**Rating:** 8
**Confidence:** 4

**Summary:**

The paper proposes a revised take of the NCA algorithm for supervised learning on tabular data, where the neighborhood aggregation is done in a representaion space of a neural network, the model is optimized via SGD and additional stochasticity is introduced in subsetting the neighbors list.

The resulting architecture is conceptually simpler than prior state-of-the-art tabular retrieval models, while improving in performance and eficiency as shown via an extensive experimental evaluation.

**Strengths:**

- The proposed method is well motivated. Attention to simple KNN-based methods in deep tabular models was limited, except TabR which is well addressed in the text.
- The method is both conceptually simple and easy to implement, without sacrificing performance
- The experimental results and ablations are extensive and insightful:
  - The step-by-step Linear-NCA ablation (table 2) is a principled and convincing way to explore a model design space
  - Stochastic Neighborhood Sampling ablation shows an interesting result (improved performance from sampling) and provides a practically important outcome for retrieval-based tabular NNs
  - Other important minor details (like the numerical feature embedding ablations), slight improvements in the distance function used, loss functions
- The writing and overall storytelling is engaging and well thought-out

**Weaknesses:**

- All experimental results and ablations rely on the average ranks of a set of methods being tested. This provides some signal for which modificaitons are usefull, but other means of comparison might make this even clearer. For example, additinal relative improvement compared to a strong baseline (e.g. a well-tuned MLP) would provide usefull additional signal besides the average rank metric (e.g. what is the scale of such improvements).
- Minor additional to the previous point: I find that providing raw unaggregated results for the core (or even for all) the experiments is very usefull for quick sanity-checks and comparisons in future work. So that others could assess results on the individual datasets by consulting the paper text.
- I believe limitation should be discussed somewhere in the main text. E.g. what are the confines of the proposed mehtod. Are there any cases where it may perform poor.
   - Maybe some post-hock meta-analysis akin to the one in tabzilla paper (https://arxiv.org/abs/2305.02997) of what are the datasets where ModernNCA performs worse.
   - Some recent benchmarks demonstrate that retrieval-based models might not be universaly superior (https://arxiv.org/abs/2406.19380)

**Questions:**

- Could you provide or point to the raw per-dataset metrics?
- Is it possible to add some other means of comparison along with average ranks (e.g. relative improvement to a baseline)? What does it show?
- What are the methods limitations?

---

> ### Author Response · Authors · 2024-11-21
> **Response (1/2)**
>
> Thanks for the valuable feedback. In this rebuttal, we address the reviewers' suggestions and concerns in a Q&A format:
>
> **Q1: Additional relative improvement compared to a strong baseline (e.g. a well-tuned MLP) would provide useful additional signal besides the average rank metric (e.g. what is the scale of such improvements)**
>
> **A1**: Thank you for the suggestion. In addition to the average rank (Figure 1), critical difference diagrams (Figure 2), and significant t-tests (Table 1), we further evaluate ModernNCA using two additional criteria to provide a clearer picture of its effectiveness:
>
> - **Shifted Geometric Mean (SGM) classification Error** and **Shifted Geometric Mean (SGM) nRMSE** for classification and regression respectively, as proposed in [A]. Here, nRMSE normalizes RMSE by dividing it by the standard deviation of the target values, making it more interpretable across datasets.
>
> $\mathrm{SGM}_{\varepsilon}:=\exp\left(\sum\limits _ {i=1}^{N _ {\mathrm{datasets}}}\frac{w _ {i}}{N _ {\mathrm{splits}}} \sum\limits _ {j=1}^{N _ {\mathrm{splits}}}\log(\mathrm{err} _ {ij}+\varepsilon)\right).$
>
>
>
>
> The results are shown below. ModernNCA achieves the best SGM values (the lower, the better) among all methods. For nRMSE, ModernNCA performs best among deep tabular methods and is only slightly outperformed by CatBoost and XGBoost in regression cases across 300 datasets.
> | **Method**           | **Classification (cls)** | **Regression (reg)** |
> |-----------------------|--------------------------|-----------------------|
> | ModernNCA            | 0.113079                | 0.375543             |
> | TabR                 | 0.116880                | 0.391278             |
> | CatBoost             | 0.125097                | 0.356144             |
> | MLP-PLR              | 0.128093                | 0.398128             |
> | XGBoost              | 0.128842                | 0.373075             |
> | FTT                  | 0.129208                | 0.399773             |
> | SAINT                | 0.132531                | 0.475766             |
> | DCNv2                | 0.133382                | 0.440412             |
> | ExcelFormer          | 0.135221                | 0.412196             |
> | MLP                  | 0.135510                | 0.430256             |
> | KNN                  | 0.159258                | 0.487707             |
> | SwitchTab            | 0.180675                | 0.981493             |
>
> - **Relative improvements over a strong baseline (MLP)**: We compute the relative improvement compared to a well-tuned MLP as proposed in [B]. We use the relative performance -1 as the final score as in [B].
>
> **The results** are listed below. ModernNCA achieves the highest relative improvement for classification tasks compared to other methods. For regression tasks, ModernNCA demonstrates significant relative improvements, performing better than other deep tabular methods and second only to CatBoost and XGBoost.
> | **Method**           | **Classification (cls)** | **Regression (reg)** |
> |-----------------------|--------------------------|-----------------------|
> | ModernNCA            | 0.030789                | 0.181178             |
> | TabR                 | 0.024230                | 0.020076             |
> | CatBoost             | 0.012500                | 0.209671             |
> | XGBoost              | 0.006976                | 0.197869             |
> | DCNv2                | 0.003726                | -1.114220            |
> | MLP-PLR              | 0.003326                | 0.017984             |
> | FTT                  | 0.002687                | 0.061669             |
> | MLP                  | 0.000000                | 0.000000             |
> | SAINT                | -0.003680               | 0.035865             |
> | ExcelFormer          | -0.009870               | 0.080631             |
> | KNN                  | -0.030300               | 0.039571             |
> | SwitchTab            | -0.065700               | -0.985630            |
>
> [A] Better by Default: Strong Pre-Tuned MLPs and Boosted Trees on Tabular Data. NeurIPS 2024.
>
> [B] TabM: Advancing Tabular Deep Learning with Parameter-Efficient Ensembling. CoRR 2024.
>
> **Q2: Providing raw unaggregated results for the core (or even for all) the experiments is very usefull for quick sanity-checks and comparisons in future work. So that others can assess results on the individual datasets by consulting the paper text.**
>
> **A2:** Thank you for the suggestion. The individual results for all datasets are provided in **Table 10** and **Table 11** in the appendix of the paper. These tables include raw, unaggregated results for both classification and regression tasks, allowing for detailed comparisons and facilitating reproducibility in future work.

---

> ### Author Response · Authors · 2024-11-21
> **Response (2/2)**
>
> **Q3: Limitation of the method. Are there any cases where it may perform poor. Maybe some post-hock meta-analysis akin to the one in tabzilla paper (https://arxiv.org/abs/2305.02997) of what are the datasets where ModernNCA performs worse.**
>
> **A3**: Inspired by Tabzilla, we conducted a meta-analysis using dataset meta-features (e.g., number of features, feature-to-sample ratio, feature sparsity, range of feature values) to train a decision tree that predicts when the performance of ModernNCA falls behind XGBoost. This analysis is based on 25 meta-features,similar to those used in Tabzilla, and identifies scenarios where ModernNCA tends to underperform compared to XGBoost. The observations are summarized below (detailed results are available in *Supplementary Material*):
>
> 1.	High Feature Sparsity: Datasets with a higher average sparsity of features pose challenges for ModernNCA.
> 2.	High Feature-to-Sample Ratio: ModernNCA struggles on datasets with many features but relatively small sample sizes, leading to a large feature-to-sample ratio.
> 3.	High Feature Skewness: Datasets with significant skewness in feature distributions (i.e., those deviating markedly from normal distributions) negatively impact ModernNCA's performance.
> This meta-analysis highlights specific dataset characteristics where ModernNCA might face limitations, providing insights for future work and potential areas for improvement. We will incorporate the discussions in the final version.
>
> **Q4: Some recent benchmarks demonstrate that retrieval-based models might not be universally superior (https://arxiv.org/abs/2406.19380)**
>
> **A4**: Thank you for pointing out this perspective. Based on our initial investigation using the TabRed benchmark, we agree that ModernNCA may lose its superiority when there is a significant distribution shift between training and test sets. This limitation arises because the prediction strategy of ModernNCA (as well as TabR) heavily relies on identifying neighbors of the test instance in the embedding space. Distributional changes can compromise the effectiveness of neighbor retrieval, making it challenging to maintain robust performance.
>
> That said, we would like to emphasize that addressing distributional shifts and improving out-of-distribution (OOD) generalization remains an open problem not only for retrieval-based models but for all kinds of machine learning methods that rely on empirical risk minimization. We plan to investigate approaches to enhance neighbor search under such conditions in the future, such as incorporating domain adaptation techniques or robust embedding learning strategies. We will also discuss this limitation and its implications in the final version of the paper.

---

> > ### Comment · Reviewer_Cjck · 2024-11-24
> >
> > Thank you for the extensive rebuttal response and new experiments! I remain positive about the submission.
> >
> > The analysis you summarize in **A3** is very insightful and usefull. Feature diversity, sparsity and skewness may also contribute to the underperformance on new benchmarks like TabReD mentioned during the discussion (the datasets in these new benchmarks have more features). The nuanced discussion of the limitations would improve the paper in my opinion.
> >
> > Importantly, I don't think that performance on new benchmarks is a significant issue for the submission, as it still provides a valuable contribution to tabular DL models based on nearsest neighbors in my view. Improving performance in the new scenarios is worthy of an extended future inverstigations. This submission should just clearly mention the limitations.

---

> > > ### Author Response · Authors · 2024-11-25
> > > **Thank you for the positive feedback**
> > >
> > > Thank you for your positive feedback following the rebuttal and for describing our analysis as "insightful and useful".  We are glad that the additional experiments and clarifications have addressed your concerns. We will incorporate the analysis into the final version of the paper to further enhance understanding of the capabilities and properties of ModernNCA.
> > >
> > > We also sincerely appreciate your recognition that this paper “provides a valuable contribution to tabular DL models based on nearest neighbors”. As suggested, we will include a clear discussion of the limitations of ModernNCA in the final version and outline potential strategies for extending its capabilities to address new settings, such as temporal split datasets in benchmarks like TabReD, in future work.

---

### Official Review · Reviewer_PJDn · 2024-11-02

**Soundness:** 3
**Presentation:** 2
**Contribution:** 3
**Rating:** 6
**Confidence:** 5

**Summary:**

This study considers learning on tabular data, and proposes ModernNCA -- a deep version of the classic Neighborhood Components Analysis algorithm. Contrary to NCA, the transformation in ModernNCA is non-linear, and is powered by a neural network. To make the training of ModernNCA more efficient and more effective, the paper also proposes Stochastic Neighborhood Sampling (SNS). In experiments on 300 datasets, ModernNCA is reported to achieve the best average rank among the considered baselines, including gradient-boosted decision trees (GBDT).

**Strengths:**

- The method is simple.
- Generally, I tend to agree that nearest neighbors may be underexplored in the context of tabular data. While TabR seems to close this gap to some extent, ModernNCA looks like a good addition to the field.
- The SNS strategy looks simple and effective, and also differentiates the method from TabR.
- On the considered benchmark, the proposed ModernNCA achieves a better average rank and a better balance of task performance and training time compared to baselines.
- A large number of baselines and datasets.
- An ablation study is provided.

**Weaknesses:**

Note: regarding the "datasets" and "metrics" weaknesses, I admit that the field lacks standardized benchmarks and metrics.

**Datasets**

My understanding is that the benchmark consists of many automatically collected datasets. In the light of the recent studies about tabular datasets, it is unclear how representative the benchmark is. Examples of the studies:

- Towards quantifying the effect of datasets for benchmarking: A look at tabular machine learning
- A Data-Centric Perspective on Evaluating Machine Learning Models for Tabular Data
- TabReD: Analyzing Pitfalls and Filling the Gaps in Tabular Deep Learning Benchmarks

**Metrics**

The metrics such as ranks or wins do not show the scale of performance gaps between methods. It is unclear how significant are the wins and losses of ModernNCA (not from statistical perspective, but from the practical perspective).

**Presentation**

- In my opinion, the presentation could be more efficient. The proposed NCA extensions are not conceptually novel, so I believe that the story on the first six pages could be more compact. Perhaps, some of the details and discussion can be moved to appendix.
- Based on my understanding of the TabR baseline
    - The explanation of TabR on L152 is not correct, since it is not a Transformer variant. Quoting the TabR paper: *"a feed-forward network with a custom k-Nearest-Neighbors-like component in the middle"*.
    - The description of TabR on L216-L226 is: (1) not complete, (2) not correct, and, if I am not mistaken, (3) not used in the story. In the light of (3), I do not go into details about (1) and (2). Perhaps, this description can be simply removed?
- Generally, communicating the empirical nature of a study as in L108-L113 is fine. However, personally, I would change the first sentence on L108 to something more neutral.
- Perhaps, Figure 1 can be placed closer to the related experiments, but this can be subjective.

**Related work.**

There is a missing related work: "Improving Generalization via Scalable Neighborhood Component Analysis" ECCV 2018. That paper also describes how to efficiently train a deep NCA, and I think their method is more advanced than the one proposed in this submission. Though their method can be too complicated for the scope of this paper. In that case, I recommend discussing this related work and explaining why the proposed SNS is a better choice for this work compared to the method from the referenced paper.

**Questions:**

-

---

> ### Public Comment · ~Yi_Ren10 · 2024-11-20
>
> The latest TabReD paper, showing that Modern NCA doesn't outperform MLP on its dataset, as shown in Table 3 and Figure1 in paper "TABRED: ANALYZING PITFALLS AND FILLING THE GAPS IN TABULAR DEEP LEARNING BENCHMARKS".
>
> Table 3:
>
> MLP rank 5
>
> ModernNCA rank 5.6
>
>
> Figure 1:
>
> relative performance percentage change over MLP is -1.05%

---

> > ### Author Response · Authors · 2024-11-23
> > **Response to Yi Ren**
> >
> > Dear Yi,
> >
> > Thanks for your interest in our paper. These two papers focus on different tabular machine learning settings.
> > The experiments in our paper are the standard tabular classification and regression tasks, where the test instances and training instances come from the same distribution. In this case, ModernNCA achieves promising performance than other comparison methods such as MLP.
> >
> > TabReD is a benchmark where the datasets are split in a temporal manner. In other words, there exists a large distribution gap between the training and test instances. As validated in TabReD paper, MLP works better than the retrieval method in this case such as TabR and ModernNCA. One possible reason is that the prediction strategy of ModernNCA (as well as TabR) heavily relies on identifying neighbors of the test instance in the embedding space. Distributional changes can compromise the effectiveness of neighbor retrieval, making it challenging to maintain robust performance.
> >
> > Addressing distributional shifts and improving out-of-distribution (OOD) generalization remains an open problem for retrieval-based models, including ModernNCA. We plan to investigate approaches to enhance neighbor search under such conditions in the future, such as incorporating domain adaptation techniques or robust embedding learning strategies. We will also discuss this limitation and its implications in the final version of the paper.

---

> > > ### Public Comment · ~Huai-Hong_Yin1 · 2025-03-18
> > > **Response to Yi Ren**
> > >
> > > Dear Yi,
> > >
> > > Thank you for pointing out the performance drop of ModernNCA in TabReD. We have updated  possible limitations of ModernNCA  in the "Limitations" section of the appendix in our final version.
> > >
> > > It is worth noting that a recent study [A] has shown that adopting alternative data-splitting protocols—such as random splits for training and validation—significantly improves ModernNCA’s performance, making it competitive with other methods. Furthermore, ModernNCA’s performance is further enhanced when incorporating temporal embeddings. This indicates that ModernNCA still has great potential in temporal shift scenarios. In future research, we will explore ways to improve ModernNCA's performance on high-dimensional data.
> > >
> > > [A] Understanding the Limits of Deep Tabular Methods with Temporal Shift

---

> ### Author Response · Authors · 2024-11-21
> **Response (1/2)**
>
> Thanks for the valuable feedback. In this rebuttal, we address the reviewers' suggestions and concerns in a Q&A format:
>
> **Q1: [Datasets] In light of the recent studies about tabular datasets, it is unclear how representative the benchmark is.**
>
> **A1**: We evaluate ModernNCA on a large-scale benchmark containing 300 datasets with sizes ranging from 1,000 to 1 million instances. This benchmark covers a diverse range of tasks, including binary classification, multi-class classification, and regression. Additionally, the datasets span various domains such as education, physics, and finance. Given the extensive scope of these datasets, we believe the improvements demonstrated by ModernNCA over other methods are robust and representative.
>
> Thank you for pointing out recent studies on tabular datasets. We will address these in the final version of the paper. Specifically:
>
> - [A] Highlights properties such as dataset count and age in evaluations. Our benchmark includes 300 datasets, exceeding the dataset counts in related studies, making our results more comprehensive.
>
> - [B] adopts a data-centric perspective to evaluate tabular datasets using 10 Kaggle datasets and emphasizes the role of feature preprocessing and engineering. We believe our study complements [B] by using a much broader dataset range while incorporating similar preprocessing techniques.
>
> - [C] focuses on datasets with latent temporal characteristics, where training and test splits have different distributions based on time. While this is not explicitly addressed in our current work, we consider this an interesting future research and extension of ModernNCA.
>
> **Q2: [Metrics] The metrics such as ranks or wins do not show the scale of performance gaps between methods. It is unclear how significant are the wins and losses of ModernNCA (not from statistical perspective, but from the practical perspective).**
>
> **A2**: In the paper, we evaluate performance using several metrics to provide a comprehensive comparison:
>
> 1. **Average Rank (Figure 1):** This metric summarizes relative performance across datasets, making it easy to interpret which method consistently outperforms others.
> 2. **Critical Difference Diagrams (Figure 2):** These are based on the Wilcoxon-Holm test with a significance level of 0.05, helping to visualize statistical differences among methods.
> 3. **Significant $t$-Test (Table 1):** This test identifies statistically significant wins or losses between ModernNCA and comparison methods at a 95% confidence interval.
>
> Regarding practical significance, win/loss ratios provide an intuitive and unbiased perspective on how often a method outperforms others across diverse datasets. These ratios are effectively “normalized” and easy to understand, offering insight into which method is a better choice on average. While performance scales (e.g., accuracy differences or RMSE gaps) could be directly analyzed, they vary significantly across datasets with different characteristics. Normalizing such metrics would make them harder to interpret, potentially obscuring practical insights.
>
> Additionally, please refer to **A1 for Reviewer Cjck**, where we include additional metrics, such as relative comparisons with the strong baseline MLP and alternative metrics like the Shifted Geometric Mean (SGM) classification error for classification and Shifted geometric mean of normalized RMSE for regression. These provide further context on performance gaps and practical significance.
>
> In summary, our metrics are designed to balance statistical rigor with practical insights, ensuring the results are interpretable and actionable for selecting the most suitable algorithm on average.
>
> **Q3: [Presentation] The proposed NCA extensions are not conceptually novel, so I believe that the story on the first six pages could be more compact. Perhaps, some of the details and discussion can be moved to appendix.**
>
> **A3**: Thank you for the suggestion. We agree that compactness can enhance readability, and we will consider moving some details from the first six pages to the appendix in the final version. We will carefully refine the presentation to balance conciseness with the importance of preserving the narrative and technical contributions that highlight the challenges and solutions involved in making NCA perform effectively on modern tabular tasks.
>
> That said, we would like to emphasize the value of the detailed exploration in the main text. Over the past two decades, despite the potential of NCA, these extensions have not been successfully applied to tabular datasets. Our paper demonstrates how vanilla NCA can be significantly enhanced with modern techniques, achieving strong results across diverse datasets. This step-by-step approach not only validates the practical utility of each extension but also provides insights that are valuable to the research community.

---

> > ### Author Response · Authors · 2024-11-21
> > **Response (2/2)**
> >
> > **Q4: [Presentation] Some descriptions of TabR.**
> >
> > **A4**: Thank you for pointing out these issues regarding the description of TabR. We appreciate the detailed feedback and will carefully revise these sections in the final version to ensure accuracy and clarity. Below are our responses to specific concerns:
> >
> > 1. L152: We say TabR is a Transformer variant since its workflow is similar to a query-key-value architecture. We will modify the description to “it employs a feed-forward network with a custom k-Nearest-Neighbors-like component”.
> > 2. L216–L226: We acknowledge that the description here just shows the main idea since the full architecture of TabR is a bit complicated. We will consider removing or significantly simplifying it to focus on the key contributions of our study.
> > 3. L108–L113: We will rephrase the first sentence on L108 to adopt a more neutral tone while preserving the empirical nature of the study.
> >
> > We greatly value the reviewer's comments and will ensure that the final version provides an accurate and concise explanation of TabR to avoid any confusion.
> >
> > **Q5: [Related Work] Discussion of the related paper [D].**
> >
> > **A5**: Thank you for highlighting the related work. We will incorporate a discussion of [D] in the final version of the paper. In a nutshell, there are several key differences between ModernNCA and [D]:
> >
> > 1. **Task Focus**: [D] addresses image classification and open-set recognition tasks, whereas ModernNCA is specifically designed for tabular data, handling both classification and regression tasks with a specially tailored feature encoder for the tabular domain.
> > 2. **Training Efficiency**: [D] scales up NCA using stop-gradient operations for partial gradient calculations and momentum updates of model parameters, where gradients are delayed. In contrast, ModernNCA accelerates training efficiency through Stochastic Neighborhood Sampling (SNS), achieving comparable training efficiency to [D] while introducing a simpler mechanism tailored to tabular datasets.
> > 3. **Memory Usage**: [D] employs an additional model for momentum updates and a memory bank to store embeddings for all training data, leading to higher memory requirements. ModernNCA, on the other hand, only uses memory approximately equal to the size of Sampled neighborhoods, making it more memory-efficient compared to [D].
> > We will emphasize the advantages of our SNS approach for tabular tasks and its effectiveness in balancing performance, efficiency, and memory usage in the discussion.
> >
> > [A] Towards quantifying the effect of datasets for benchmarking: A look at tabular machine learning
> >
> > [B] A Data-Centric Perspective on Evaluating Machine Learning Models for Tabular Data
> >
> > [C] TabReD: Analyzing Pitfalls and Filling the Gaps in Tabular Deep Learning Benchmarks
> >
> > [D] Improving Generalization via Scalable Neighborhood Component Analysis

---

> > > ### Comment · Reviewer_PJDn · 2024-11-23
> > >
> > > I thank the authors for the rebuttal. I would like to keep my score. For me, what I marked as "strengths" still outweighs what I marked as "weaknesses".
> > >
> > > **Regarding the datasets,** I would like to clarify my point. I agree that the number of datasets is a strength of the used benchmark, as well as the ambition to cover many domains. My concerns are about the *quality* of these datasets, and, as a consequence, about the reliability of the obtained conclusions. Just from a quick look at the used datasets (introduced in "A Closer Look at Deep Learning on Tabular Data"):
> > > - Extensive use of "multi-version" datasets, i.e. datasets that are closely related, or are even obtained from the same data. This weakens the argument about the number of datasets, and introduces bias towards these datasets. Examples: `BNG-*` datasets, `Contaminant-detection-*` datasets, `FOREX-*` datasets, any many other such cases.
> > > - Use of datasets with label leakage. Some of the datasets affected by this issues were identified only in recent work such as "TabReD: Analyzing Pitfalls and Filling the Gaps in Tabular Deep Learning Benchmarks", so I don't position it as a critical overlook of this submission, but it still illustrates my point.
> > > - Use of trivial datasets solvable with the perfect quality (i.e. 1.0 accuracy).
> > > - Etc.
> > >
> > > Things like the above are a natural consequence of using public datasets (e.g. from OpenML) without much filtering. And my point is that the formal number of datasets and covered domains cannot be the only perspective on the benchmark. In particular, as I see it, the studies I mentioned motivate investing more in the dataset filtering.
> > >
> > > **Regarding the metrics,** I thank the authors for computing new metrics.

---

> > > > ### Author Response · Authors · 2024-11-25
> > > > **Thanks for the  positive feedback**
> > > >
> > > > Thank you so much for your timely feedback. We are glad that the aspects the reviewer highlighted as "strengths" still outweigh those referred to as "weaknesses."
> > > >
> > > >
> > > > Regarding the dataset concern, we would appreciate your further clarification. We agree with you that besides “diversity”, “quality” is also an important factor in datasets, and we humbly think this is a common issue applicable to most if not all of the tabular data papers. While smaller benchmarks (e.g., with tens of datasets) are relatively easy to have quality control, there is a force in the tabular data community to study larger, more diverse benchmarks, on which quality control becomes harder. We think a dedicated, systematic study of how to construct a large yet high-quality benchmark is needed in the tabular data community and deserves a separate paper (or even a journal article with more pages).
> > > >
> > > > For our current manuscript, we focus on methodology like many existing tabular data papers. That said, we will add a clarification/discussion paragraph in the final version describing the potential problems in the evaluation data. We also plan to conduct a more formal study in the future following your suggestion about dataset filtering.

---

### Meta-Review · Area_Chair_RgHS · 2024-12-12

**Metareview:**

This paper explores more traditional approaches for supervised tabular data modelling integrated with modern deep learning techniques. It uses a Neighbourhood Components Analysis approach in the latent space of a neural network which can be optimized with SGD.

Reviewers brought up many concerns, but most of them were addressed during the discussion. The primary outstanding concern from several reviewers, but not all, was around novelty, in that this work revives old techniques through a modern lens. In my view this approach is sufficiently novel, at least compared to the typical paper accepted at ICLR, and considering the positive empirical results I am recommending acceptance as a Poster.

Several reviewers asked about the limitations of this method, and cases where it may underperform. The authors provided some useful thoughts, which should be featured in the main text of the paper so that readers are properly informed. I request that the authors make this change for the camera-ready version.

**Additional Comments On Reviewer Discussion:**

The main points of concern were: novelty - the method is a combination of old and new techniques (but the authors were up front about this in their original submission); lack of insights, theoretical or otherwise, into why this method works or cases when it does not; reliance on ranking metrics which may hide the scale of outperformance or significance of results; choice of Euclidean distance only; sensitivity to hyperparameters; and other minor concerns.

During the discussion the authors addressed the reliance on ranking metrics by including raw metrics and reviewers were able to weigh in on whether they found the performance gains significant enough. The other points on choice of Euclidean distance, hyperparameters, etc., were discussed and largely resolved. Novelty was the main outstanding point for some but not all reviewers as mentioned in the metareview.

---

### Decision · Program_Chairs · 2025-01-22

Accept (Poster)